



# First implementation of a new cross-disciplinary observation strategy for heavy precipitation events from formation to flooding

Andreas Wieser[1*], Andreas Güntner[2,8*], Peter Dietrich[3,9], Jan Handwerker[1], Dina Khordakova[4], Uta Ködel[3], Martin Kohler[1], Hannes Mollenhauer[3], Bernhard Mühr[5], Erik Nixdorf[6,10], Marvin Reich[2], Christian Rolf[4], Martin Schrön[3], Claudia Schütze[7], Ute Weber[7]

[1] Institute for Meteorology and Climate Research, Karlsruhe Institute of Technology (KIT), Karlsruhe, 76021, Germany
[2] Helmholtz-Centre Potsdam GFZ German Research Centre for Geosciences, Potsdam, 14473, Germany
[3] Department Monitoring and Exploration Technologies, Helmholtz Centre for Environmental Research (UFZ), Leipzig, 04318, Germany
[4] Institute of Energy and Climate Research (IEK-7), Forschungszentrum Jülich (FZJ), Jülich, 52428, Germany
[5] EWB Wetterberatung, Karlsruhe, 76199, Germany
[6] Department Environmental Informatics, Helmholtz Centre for Environmental Research (UFZ), Leipzig, 04318, Germany
[7] Department Computational Hydrosystems, Helmholtz Centre for Environmental Research (UFZ), Leipzig, 04318, Germany
[8] Institute of Environmental Science and Geography, Potsdam University, Potsdam, 14476, Germany
[9] Center for Applied Geoscience, University of Tübingen, Tübingen, 72074, Germany
[10] Federal Institute for Geosciences and Natural Resources, Department of Groundwater and Soil, Hannover, 30655, Germany
* These authors contributed equally to this work.

*Correspondence to*: Andreas Wieser (andreas.wieser@kit.edu) and Andreas Güntner (andreas.guentner@gfz-potsdam.de)

If available, the 16-digit ORCID of the author(s):

Andreas Wieser: 0000-0003-1266-5432
Andreas Güntner: 0000-0001-6233-8478
Peter Dietrich: 0000-0003-2699-2354
Jan Handwerker: n/a
Dina Khordakova: 0000-0002-4331-2288
Uta Ködel: 0000-0002-0296-6250
Martin Kohler: 0000-0003-0928-8314
Hannes Mollenhauer: 0000-0002-4746-9143
Bernhard Mühr: n/a
Erik Nixdorf: 0000-0001-5569-4597
Marvin Reich: 0000-0001-7301-2094
Christian Rolf: 0000-0001-5329-0054
Martin Schrön: 0000-0002-0220-0677
Claudia Schütze: 0000-0001-9562-8965
Ute Weber: n/a





**Abstract**:

Heavy Precipitation Events (HPE) are the result of enormous quantities of water vapour being transported to a limited area.
HPE rainfall rates and volumes cannot not be fully stored on and below the land surface, often leading to floods with short forecast lead times that may cause damage to humans, properties, and infrastructure. Towards an improved scientific understanding of the entire process chain from HPE formation to flooding at the catchment scale, we propose an elaborated event-triggered observation concept. It combines flexible mobile observing systems out of the fields of meteorology, hydrology and geophysics with stationary networks to capture atmospheric transport processes, heterogeneous precipitation patterns, land
surface and subsurface storage processes, and runoff dynamics.

As part of the Helmholtz Research Infrastructure MOSES (Modular Observation Solutions for Earth Systems), the added value of our observation strategy is exemplarily shown by its first implementation in the Mueglitz river basin (210 km²), a headwater catchment of the Elbe in the Eastern Ore Mountains with historical and recent extreme flood events. Punctual radiosonde observations combined with continuous microwave radiometer measurements and back trajectory calculations
deliver information about the moisture sources, initiation and development of HPE X-Band radar observations calibrated by ground based disdrometers and rain gauges deliver precipitation information with high spatial resolution. Runoff measurements in small sub-catchments complement the discharge times series of the operational network of gauging stations. Closing the catchment water balance at the HPE scale, however, is still challenging. While evapotranspiration is of less importance when studying short term convective HPE, information on the spatial distribution and on temporal variations of
soil moisture and total water storage by stationary and roving cosmic ray measurements and by hybrid terrestrial gravimetry offer prospects for improved quantification of the storage term of the water balance equation. Overall, the cross-disciplinary observation strategy presented here opens up new ways towards an integrative and scale-bridging understanding of event dynamics.

**1.       Introduction**

In the past two decades an increase in and intensification of extreme weather events was observed, which exerted considerable impact on, for example, food production, public health, water and air pollution (Bastos et al., 2020). It is mainly the socio-economic impact that acts as a catalyst for research on the importance of such distinct dynamic events for long-term global change processes. Blöschl et al. (2019) identified several major heretofore unsolved scientific questions focused on the process-
based understanding of hydrological variability and causality. As one of the most urgent tasks in hydrological research for the coming years, they highlighted that a new focus is required for a complete understanding of how environmental change propagates across interfaces within the hydrological system. These interfaces are those between compartments (e.g. atmosphere–vegetation–soil–bedrock–streamflow–hydraulic structures) and those between processes that are usually dealt with by different disciplines.





The increase in intensity and occurrence of Heavy Precipitation Events (HPE) due to climate change is well documented in observational records and represented in climate models (Fisher & Knutti, 2015). However, scientific understanding of the entire process chain that propagates HPE – from the different terms of the catchment water balance to the flood generation and the respective impact – is still lacking (Myhre et al., 2019). In particular, this understanding is often limited by a gap in observational data, their unsatisfactorily coarse spatial and / or temporal resolution, the inability to consistently close energy and water balances through observations, and by missing monitoring techniques that would facilitate direct quantification of the entire water flux and storage processes from the atmosphere through the surface of the Earth down to the groundwater. The resulting lack of event understanding also hinders improvement of modeling approaches which in turn would benefit from extended observation data sets both for model validation and data assimilation towards better prediction capabilities.

Environmental research infrastructures (RI) are designed to observe a wide range of biotic and physical processes linking atmosphere, biosphere, hydrosphere and geosphere (Chabbi et al., 2017). Due to the complex responses and feedback involved in environmental processes, their successful investigation requires an integrated and cross-compartmental approach. Established terrestrial RIs are typically designed to provide long-term, consistent and standardized observations of environmental processes, their causes and their interactions with regard to climate change and global warming (e.g. ICOS, TERENO, FLUXNET, ACTRIS). Hence, they mostly operate as stationary observatories at a certain location of interest.

To complement and extend these observatories, the Modular Observation Solutions for Earth Systems (MOSES) RI has been established as a mobile research facility for short term observations of extreme events. It builds on such existing monitoring networks serving as anchor points or baseline data source to analyze the impacts of extreme events (Weber et al., 2022). However, a mobile RI like MOSES has to face event-driven campaigns with an essential requirement: to be in the right place at the right time. Its operation demands thorough preparation with respect to suited localities and observational periods, which is based on historical data, current seasonal developments and short-term forecasts during the entire measuring campaign period. High flexibility and foresighted planning with regard to measurement systems and available personnel are prerequisites, including the setup of appropriate observation sites, installation of the measuring systems, and provision of fast and reliable data transmission to control and synchronize the ongoing measurements. This desired flexibility guided the design of MOSES with modular deployable measuring systems to cover the challenge of cross-compartmental process monitoring (Weber et al., 2022).

Here we present the design of the MOSES mobile research facility for investigations along the hydro-meteorological event chain from HPE to floods, and the first implementation of our observation strategy in the Mueglitz river basin (210 km²), a headwater catchment of the Elbe in the Eastern Ore Mountains in 2019. By means of three examples we illustrate (i) measurement synergies combined with model trajectories to delineate moisture variations in the atmosphere and track it back to its sources, (ii) calibrated high resolution precipitation data for the entire catchment to understand the spatial soil moisture patterns and runoff generation, and (iii) a new measurement system based on terrestrial gravimetry to determine changes in



total water storage. The 2019 Mueglitz campaign was the first deployment of this mobile observation platform during the MOSES implementation phase from 2017 until 2021.

## 2.    Event Driven Observation Design

Successful event monitoring relies on both adapted and flexible observation and operation concepts as well as on the technological development of mobile scientific field equipment. Our observing strategies for event-driven field campaigns are based on experience in trans-disciplinary long-term environmental monitoring such as TERENO (Zacharias et al., 2011) overlapping different compartments (atmosphere, geosphere, aquatic systems and biota) as well as on research activities that included short-term field campaigns with accompanying model studies e.g., HyMeX (Drobinski et al., 2014), DESERVE

(Kottmeier et. al., 2016) or COPS (Kottmeier et al., 2008, Wulfmeyer et al., 2011).

For the MOSES research infrastructure, three different deployment concepts adapted to the event type in focus have been developed: I) Long-term planning campaigns are required for remote and highly international observation activities. II) Medium-term planning campaigns with about one year preparation time are most suitable to set up targeted weather extremes campaigns in the European Union. III) Ad-hoc campaigns with a lead time of a few days to hours have been developed to

directly capture an event as it passes through the study area. The investigation of Heavy Precipitation Events (HPE) in the Mueglitz river basin was based on concept II in combination with III.

Such investigation of hydro-meteorological extremes provides unique opportunities to study the deflection and resilience behavior of the hydrological processes and the water balance of an entire catchment, as well as associated hydrological risks (Blöschl et al., 2013; Kreibich et al., 2017). For a holistic approach aiming at the understanding of hydro-meteorological

events, the identification of links between meteorological events, processes related to land-surface / atmosphere interaction in terms of moisture and energy fluxes, and subsurface storage mechanisms have to be taken into account (Fig. 1). This requires a cross-compartmental and multi-disciplinary research strategy, such as provided by MOSES.





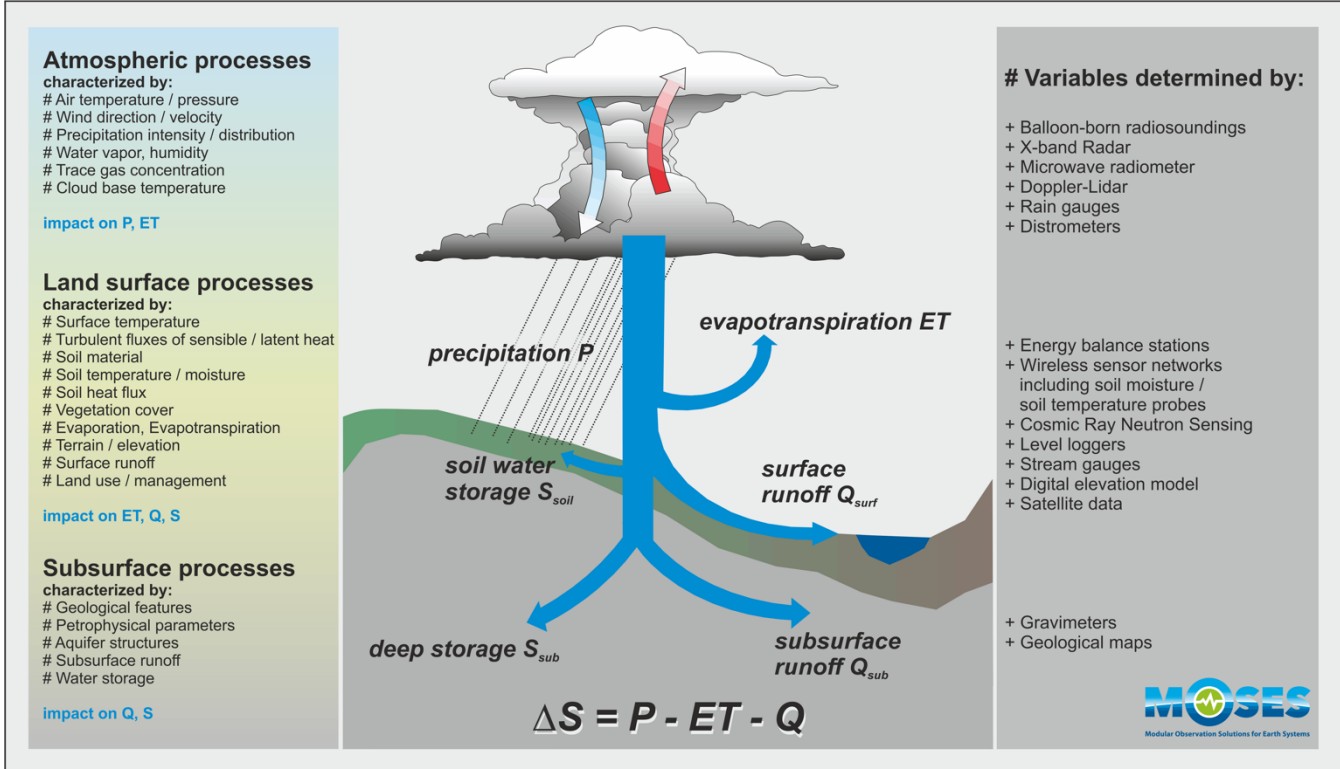

**Figure 1:  Schematic overview of the water balance components, involved processes and related variables and observation systems.**


## 2.1    Observation strategies for heavy precipitation and flood events

Flood events have forecast periods of a few days when they are related to large-scale atmospheric low-pressure systems and only a few hours when they are generated by local-scale deep convective HPE. In general, these convective HPE have a
dedicated seasonal occurrence (see section 3.2). Therefore, the best way to address the observational challenges is to combine the organizational monitoring approaches of medium-term planning campaigns covering the period of highest occurrence, which was May - July/August in the case of the Mueglitz catchment, with ad-hoc campaign components when HPE were approached:

(1) The medium-term planning campaign concept covered the observation period from May until August 2019 as a fully-
equipped campaign with highest performance in terms of data acquisition. The full range of equipment including large and less mobile measurement facilities such as the KITcube observatory (Kalthoff et al., 2013) or the gPhone solar cube (see section 3.5) were deployed. After comprehensive pre-site surveys beforehand, the equipment was installed as a temporary, distributed observatory in the Mueglitz catchment for an extended observation period (EOP) of four months during the main



event season. Permanent, mostly automatic or remote-controlled monitoring of a set of pre-defined environmental parameters
was conducted and data was processed and transferred in near real time according to defined procedures. During daily
briefings, the campaign team on duty analyzed the current weather forecast for upcoming HPE, which were then specifically
investigated by ad-hoc intensive operation phases (IOP). A decision matrix merges forecasted precipitation type and amount,
it's probability and the overall suitability of the weather situation for the expected event (Tab. 1). Together with information
on the equipment's status and the availability of staff and accommodation, the decision to announce, continue or finish an IOP
was taken by noon each day and immediately made available by email and messenger services. During an IOP all instruments
are activated at the main site but also in a wider surrounding in order to conduct pre-defined coordinated measurements with
highest resolution according to the anticipated development. Additional staff is sent out to perform, on site, manually controlled
measurements such as radiosoundings.

(2) The above mentioned IOPs correspond to the ad-hoc campaign concept, which uses the most mobile and flexible devices
for rapid operation (e.g. autonomous mobile meteorological stations, roving Cosmic Ray Neutron Sensing (CRNS), discharge
measurements, field gravimeters). The equipment and staff thereby had to be on standby and ready to start the mobile
measurements in an event case with short lead time, providing short-term event data. For this purpose, potential time frames
and target regions suitable for operation were pre-defined. During the Mueglitz campaign of 2019, the observational periods
and regions for (1) and (2) were identical, but single ad-hoc campaigns can also be performed independently.

Based on the two monitoring options described above, all components of the instrumentation were evaluated in terms of their
suitability for event-driven monitoring and their operating expense. This information is summarized in Table 2. In particular,
the personnel and time requirements as well as the mobility of the devices are crucial for campaign planning.

**Table 1: Decision matrix to judge the weather situation up to 96 hours in advance for possible HPE and IOP suitability. For a fictitious weather development, the matrix shows the forecasted precipitation type and amount and it's expected flood impact for**
**the selected catchment. Color codes according to predefined threshold values indicate the alert status from green (no or less precipitation expected / unsuitable for IOP run) to red (HPE and flooding most likely / ideal IOP case). According to the matrix scheme, decisions concerning IOP performance become clear and easily available for all involved campaign teams.**

| Day | | | today | +24 … +48 h | +48 … +72 h | +72 … +96 h |
|---|---|---|---|---|---|---|
| Precipitation | Large-scale, stratiform | | 0 | 0 | < 1 mm | 1 … 10 mm |
| | Large-scale, convective | | 0 | 0 | 10 … 40 mm | 0 |
| | Local-scale, convective | | 0 | 5 … 30 mm | 0 | 0 |
| Flood impact | | | none | low | medium | low |
| Forecast quality | | | very high | medium | very high | high |
| **Suitability for IOP** | | | **unsuitable** | **possible** | **ideal** | **unsuitable** |



## 2.2 Observation Methods

To determine the water balance components according to Fig. 1 microwave radiometers, Doppler lidar and radio soundings provide information on the state of the atmosphere as well as changes in water vapor distribution and content, temperature and wind profiles from the surface over the planetary boundary layer up into the lower stratosphere. High resolution precipitation intensity and areal distribution is derived from X-Band radar measurements which are calibrated by optical disdrometers and institutional rain gauges network data.

Land surface processes like the surface runoff variability is determined by level measurements and stream gauges in the main river and its tributaries. The evapotranspiration is quantified using energy balance stations according to Mauder and Foken (2011). Soil moisture variability indicates the shallow storage processes and is determined on a larger scale by roving CRNS and on local scale by stationary CRNS, and in situ soil moisture sensor networks.

The determination of variations in subsurface storage term can be derived by non-invasive geophysical methods, among which
terrestrial gravimetry can provide integrative information on the total subsurface storage variations.

**Table 2: Methods as part of the HPE observation strategy from rainfall formation to flood generation. The time required for pre-event site selections is not included in the installation effort.**

| Compart-ment | Target variable | Method | Resolution | Footprint | Installa-tion effort | Mobility |
|---|---|---|---|---|---|---|
| Atmo-spheric processes | **Precipitation** (distribution, intensity) | Mobile X-Band radar | 300 m / 1°, volume scan every 300 s | 100 km range | 2 persons, 1 day requires free visibility | **medium…low**, mounted on a trailer with internal generator, installation on seatainer stack with grid power preferable |
| | **Precipitation** (intensity, drop size) | Optical distrometer | 20 cm, 15 s | few meters | 1 person, 1 hour | **medium,** stand-alone operation in field |
| | **Atmospheric state** (p, T, water vapor, ozone, cloud base | Microwave radiometer | 50 … 500 m vertical, 1.5 s | Altitude range 2 m … 10 km | 2 persons, 0.5 day | **medium,** best with KITcube base camp grid power needed, liquid nitrogen for calibration |





| | | | | | |
|---|---|---|---|---|---|
| | height, liquid water path) | Radio-sounding | 5 m vertical, 1 h | Altitude range 20 m … >20 km | 2 persons, 0.5 day | **medium,** requires a base camp (KITcube) with helium storage for balloon filling |
| | **Wind profiles** | Doppler lidar | 25 m vertical, 15 s | Altitude range 50 … 3175 m | 2 persons, 0.5 day | **medium,** requires grid power or daily generator refuelling |
| **Land surface processes** | **Energy balance** (p, T, wind speed and direction, shortwave and longwave radiation budget, precipitation, soil moisture, soil temperature, surface temperature, turbulent fluxes) | Energy balance station | 10 min resp. 30 min | 400 m | 2 persons, 0.5 day | **medium,** stand-alone operation, need prerequisites of the installation site |
| | **Soil moisture** | Soil moisture wireless sensor network | 0.1 m, minutes | 1 m | 1 person, 2 hours per node (depen-ding on subsur-face con-ditions) | **medium,** standing time after installation in soil required, stationary operation after standing time |
| | | Cosmic ray station | 200 m, hourly | footprint radius 200 m | 1 person, 2 hours | **medium,** station ary operation after installation |
| | | Cosmic ray rovering | 100 m, minutes | footprint radius 200 m, spatial extent several 10 km | 1 person per monito-ring day | **high,** system is installed on all-terrain vehicle |
| | **Surface runoff** | Level logger | 0.05 m, seconds … minutes | several 100 m | 1 person, 1 day | **medium,** requires sufficient |





| | | | | | | riverbed properties to fasten the data loggers and river bank accessibility |
|---|---|---|---|---|---|---|
| **Subsurface Processes** | **Total subsurface water storage variations** | Terrestrial gravimetry: time-continuous | 100 m, minutes | 100 m | 2 persons, 2 days | **low,** solar cube (container) deployment in field |
| | **Total subsurface water storage variations** | Terrestrial gravimetry: time-lapse network survey | depending on density of network points, 1 day | footprint 100 m, spatial extent several 10 km | 1 person per monito-ring day | **high,** field vehicle |

An overview of all applied methods, instrument types deployed and their characteristic requirements concerning their
application for event observations is listed in Table 2, from which the following methods are of key importance for our
observation concept:

**Balloon sounding**

Standard radiosoundings using Vaisala RS41-SGP and GRAW DFM-09 sondes provide in situ measured vertical profiles of
temperature, pressure, humidity and winds up to 20 km altitude. The measurement data is transmitted via radio connection to
the ground station. Both sonde types provide high quality measurements, correspond to the Global Climate Observing System
(GCOS) Reference Upper-Air Network (GRUAN) standards and are evaluated by the World Meteorological Organization
(WMO) radiosonde intercomparison (WMO, 2011) in case of the DFM-09, or individually for the new RS41 (Khordakova et
al., 2021, Rosoldi et al., 2021, Jing et al., 2021). In addition to the standard radiosondes, larger balloons with enhanced
instrumentation were launched in order to measure tropospheric moist air mass transport into the dry lower stratosphere by
overshooting convection. This instrumentation consists of an electrochemical concentration cell (ECC) ozone sonde (Smit et
al., 2007), and a cryogenic frost point hygrometer (CFH; Vömel et al., 2007; 2016). The CFH is well suited for measuring low
water vapor concentrations prevailing in the upper troposphere and lower stratosphere where standard radiosondes are less
accurate. Beside these trace gas instruments, aerosol and cloud particles are measured with the backscatter sonde COBALD
(Brabec et al., 2012). Overshooting convection was observed in IOP4 during the 2019 Mueglitz campaign with this
instrumentation and analyzed by Khordakova et al. (2022).





## Microwave Radiometer

A scanning microwave radiometer RPG HATPRO-G4 (Humidity And Temperature PROfiler) measured continuously the atmospheric emissions of liquid water, water vapor and oxygen. The measurements were carried out at 14 frequencies in two frequency bands, the K-Band ranging from 22.24 to 31.40 GHz, and V-Band from 51.26 to 58.00 GHz. Using a retrieval
algorithm provided by the University of Cologne (Crewell and Löhnert, 2003; Löhnert et al., 2009) humidity profiles, integrated water vapor (IWV) and liquid water path (LWP) were obtained from the measured emission in the K-Band, whereas temperature profiles are retrieved from the V-Band.

## Doppler lidar

Continuous wind profiles covering the boundary layer including sufficient aerosols serving as scatterers are measured by a
Leosphere scanning Windcube WLS200s. The system uses invisible eye-safe laser pulses of 1.54 μm which are directed in the upper hemisphere by a two-axis scanner. Assuming that the scatterers have negligible fall speed the instrument measures radial (along beam) wind velocity for 126 range gates of 25 m length. Applying the Doppler beam swinging (DBS) technique (Lundquist et al., 2015) in a 5 point stop and stare scan at 75 deg. elevation wind profiles are derived from 50 to 3175 m altitude with a so-called display resolution of 25 m every 15 s.

**Rain radar**

Area-wide precipitation is determined by an X-Band radar (Meteor50DX, Leonardo, Neuss). Radar is able to cover large areas (even volumes) at high resolution without leaving gaps, but the physically measured value is reflectivity and not rain intensity. The latter is derived by a rule of thumb, the so-called Z/R-relation $Z=aR^b$ with reflectivity Z in $mm^6$ $m^{-3}$, R rain intensity in $mm$ $h^{-1}$ (Sauvageot, 1994).
Only a multi-step procedure is able to deliver the required reliable precipitation intensities. In a first step, interfering signals are removed by a fuzzy logic algorithm which relies on measured values of reflectivity, radial velocity, differential reflectivity as well as the texture of the polarimetric measures differential reflectivity, copolar correlation coefficient, and differential phase (Krause, 2016). Then the attenuation caused by precipitation is corrected by the ZPHI algorithm (Testud et al., 2000). This algorithm relies on differential reflectivity as a measure for the total attenuation along a radar beam. It then "distributes"
the attenuation according to reflectivity. The third step then corrects for the additional attenuation caused by rain on the radome by a simple, linear approach, assuming the (logarithmic) attenuation A (in dB) to be linearly dependent on the rain rate R (in $mm$ $h^{-1}$), measured by the collocated Parsivel disdrometer (see below): A=0.17*R. After these corrections a final calibration is applied. Since the absolute calibration of a radar is difficult and for a mobile device it is even more difficult, the total precipitation amount determined by the radar after steps 1 to 3 is therefore compared to the in situ rain measurements of the



rain gauge network operated by German Weather Service (DWD). Based on 45 stations in the Mueglitz area a single calibration factor of 1.5 dB was calculated and applied for all measurements within this campaign.

The rain intensity is then derived using the lowest, undisturbed measurement (based on a digital elevation model) of the radar volume scan.

**Disdrometer**

In contrast to rain gauges which only measure precipitation intensities, disdrometers are able to determine the size of individual precipitation particles. The applied optical disdrometer "Parsivel" (PARticle SIze and VElocity, Löffler-Mang and Joss, 2000) additionally determines the sedimentation velocity of each particle. The detection is based on a horizontal laser band of 28 mm width and 1 mm height. The measurement area is 50 cm$^2$ between a laser diode with beam forming optics on the sending, and focusing optics on the detector side.

Each hydrometeor (water droplet, snow or ice particle) falling through the laser band causes an attenuation of the received signal. The particle size is determined by the maximum attenuation and the particle fall velocity by the duration of attenuation. This allows the device to detect the hydrometeor type, apply a typical density and calculate the water equivalent precipitation rate.

Hydrometeors detected during a preset measuring interval (15 s) are aggregated in 32 size and 32 velocity classes resulting in

a 2D frequency distribution. For evaluation purposes, drop size distribution should contain a minimum of 100 drops (Handwerker and Straub, 2011). To this end, measurements are aggregated for 60 s or longer periods.

**Energy Balance Station**

The energy balance station is a multi-device platform that provides in situ measurements of standard meteorological variables like air temperature, surface temperature, relative humidity, air pressure, wind speed, wind direction and precipitation enhanced

by soil moisture and soil temperature profiles. Soil moisture was measured with devices based on Frequency Domain Reflectometry (FDR) and were installed horizontally in four depths. Sensor probes and installation configurations are comparable to the setup of the soil moisture sensor network (next paragraph).

Additionally, the short wave and long wave components of the radiation budget are measured. Standard variables and radiation components are sampled with 1 Hz, the data is then aggregated to 10 min means.

To obtain the energy balance near the surface the fluctuations of air temperature and absolute humidity are measured with a sonic anemometer and a fast open path $H_2O/CO_2$ gas analyzer (LICOR 7500) at a sampling frequency of 20 Hz. From the measurements the turbulent fluxes of latent and sensible heat are derived as 30 min averages using the eddy covariance method. The soil heat flux is measured with soil heat flux plates. Detailed information about the instruments can be found in Kalthoff et al., 2013.





**Mesoscale soil moisture monitoring**

The mesoscale soil moisture monitoring was done with an in situ wireless sensor network (WSN) using soil moisture probes (Frequency Domain Reflectometry FDR sensors,) installed in different soil horizons and the CRNS. The in situ system is based on an indirect measurement of soil moisture using the relationship between measured dielectric permittivity and liquid soil water content and includes standard SMT100 probes (Truebner GmbH, Neustadt, Germany, Bogena et al., 2017). The measuring principle and calibration procedure correspond to the explanations of Fersch et al. (2020). The soil moisture values in $m^3$ $m^{-3}$ were derived from the measured permittivity values using the empirical transfer function of Topp et al. (1980). The integration depth of CRNS monitoring covers about the uppermost 0.3 m of the soil (Schrön et al., 2017).

The CRNS as an emerging technology was applied to close the scaling gap between in situ point measurements and satellite-derived remote sensing data products (e.g., from SMOS, SMAP sensors, Andreasen et al., 2017). CRNS estimates the area-average water content by counting the cosmic ray neutrons in the air with a horizontal footprint of hundreds of meters and a vertical footprint of tens of centimeters in the soil (Köhli et al., 2015). There are two deployment modes investigating the above-ground neutron flux: (1) a stationary installation of CRNS detectors to obtain time series of temporal soil moisture variability in the footprint area (Zreda et al., 2012, Schrön et al., 2018(b)) and (2) the operation as CRNS rover based on the detector installation on an all-terrain vehicle (ATV) to acquire the spatial soil moisture distribution on a larger scale (McJannet et al., 2017, Schrön et al., 2018(a), Vather et al., 2019, Schrön et al., 2021).

**Terrestrial gravimetry**

For the monitoring of water storage changes in the sub-surface at depths larger than what is accessible by the soil moisture sensor networks and CRNS, we include the method of terrestrial gravimetry in our monitoring setup. The details of the measurement principles and the network design are specified in Subsection 3.5.

**Discharge measurement**

LTC (Level-Temperature-Conductivity) data loggers (Solinst Canada Ltd., Georgetown, ON, Canada, Toran, 2016) are small waterproof devices recording water level, temperature, and conductivity with 5-minute temporal resolution (Hannemann et al, 2022). These low-cost loggers have been installed to monitor the surface water dynamics in 5 small streams that drain sub-catchments around the main site, close to their confluence with a higher-order stream (Fig. 3). Level measurements were corrected by air pressure by using a baro-logger (Solinst Canada Ltd., Georgetown, ON, Canada) which logs changes in atmospheric pressure and temperature. The records of the electric conductivity channel were used to remove periods of data logger exposure to air from the derived dataset. Discharge time series are obtained from the water level time series by applying the Manning formula (Manning, 1890) on each channel profile using manual discharge measurements to calibrate Manning's Roughness Coefficient.



Figure 2 gives an overview of the different data processing and interpretation workflows. The interdisciplinary approach helps to combine and jointly analyze the different strands of the event chain and consistently links the meteorological events (HPE) with the hydrological impacts such as soil moisture response, runoff development on an entire catchment system. A detailed discussion of examples of such data streams will be considered in the following section.


**Figure 2: Chart of the interdisciplinary data processing and analysis workflow to link HPE with the impact processes in a catchment system considering the different terms of the water balance. The runoff is the target value with regard to floods.**




## 3.    Implementation and first case study: Mueglitz catchment 2019

### 3.1    Measurement Area

The Mueglitz river has its source at 905 m amsl on the Ore mountains ridge at the border to the Czech Republic and discharges into the Elbe in Heidenau at approx. 110 m amsl (Fig. 3). The river has a total length of about 49 km with a catchment size of

210 km². The catchment area consists of narrow, populated valleys that have only a few natural retention areas. More than half of the total area is used for agriculture, with a grassland proportion that is slightly higher than the arable land. The forest percentage share of 36% is concentrated on the unfavorable locations such as steep valley slopes. Paragneisses largely dominate the geological structure in the high altitudes with striking basalt domes, such as the Geisingberg (832 m amsl). Geomorphologically, wide-stretched rolling plateaus dominate the area intersected by narrow V-shaped and narrow U-shaped

valleys (Gerber, 2008). It is reported that due to heavy precipitation events from 1609-2020, 18 severe flood disasters happened (Walther & Pohl, 2004). In 1897, 1927, 1957, and 2002 several flood events occurred due to extreme precipitation with more than 150 mm within a few hours with catastrophic dimensions. In August 2002, heavy precipitation with more than 300 mm precipitation within 24 hours (Ulbrich et al., 2003) was recorded. At that time, this extreme amount was the highest daily precipitation volume ever measured in Germany (DWD, 2020). The storage capacity of the soil was reached and more than

50% of the precipitation resulted in runoff, leading to a peak discharge of about 400 m$^3$ s$^{-1}$ at the most downstream gauging station in Dohna (Fig. 3), compared to a mean discharge of 2.5 m$^3$ s$^{-1}$. A resulting dam break caused a flash flood with massive destruction of the infrastructures along the entire Mueglitz catchment.

The main site of this HPE monitoring campaign with the KITcube (564 m amsl) and the gPhone Solar Cube (568 m amsl) is located about five kilometers southeast of the small town Glashuette on a plateau to the east of the Mueglitz valley. The

surroundings of the site are predominantly used for intensive agriculture. The terrain descends by approximately 20 m to the east, west, and south. The soil is mainly brown soil (BBn) and Gley-Pseudogley (GG-SS) in the northeastern plateau. Near-surface geophysical data (EMI, Gamma-ray) taken in 2020 show a nearly homogeneous distribution of physical soil characteristics in the upper layers around the main site. The aquifer is located in metamorphic solid rock, and the groundwater level is deeper than 6 m. The remote sensing devices of the KITcube were installed close to a wind turbine site. To ensure the

representativeness of the near surface measurements, the in situ devices were placed in a corn field approx. 80 m west (upwind). The installation was done right after sowing and the corn reached the blossom phase in July during the teardown. An additional site, equipped with an energy balance station and a Parsivel was located near the mountain crest at Zinnwald (877 m amsl).



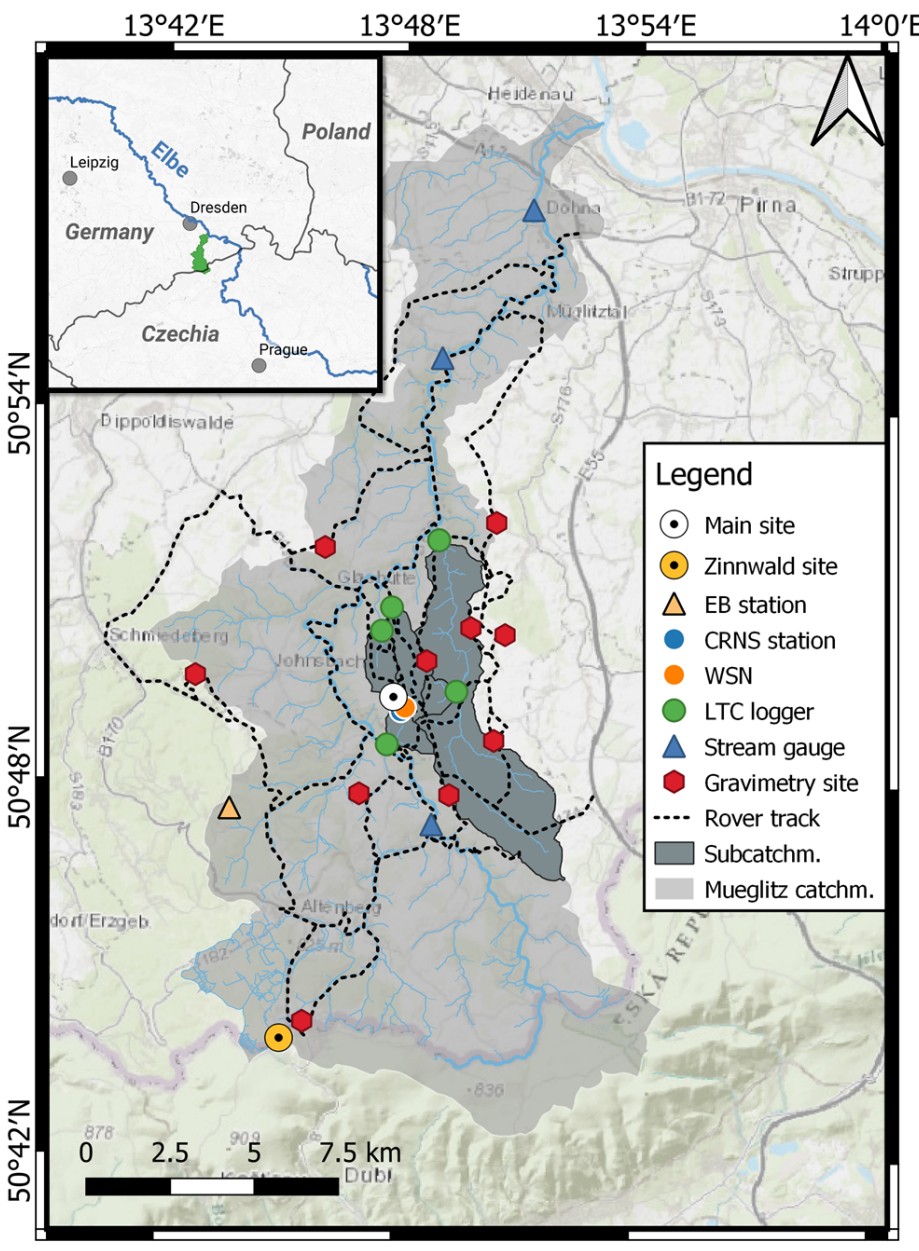

**Figure 3: Geographical location of the study area and monitoring points in the Mueglitz catchment. The upper-left plot visualizes the location of the Mueglitz catchment within Central Europe. Field gravimeter sites (red hexagons) and rover tracks with CRNS sensors (dashed lines) were operated during selected IOPs only. Abbreviations: EB: Energy Balance; WSN: Wireless Soil moisture Network; LTC Level-Temperature-Conductivity; CRNS: Cosmic Ray Neutron Sensing. Base map sources: Esri, CGIAR, USGS|GeiSN, GUGGiK, HERE, Garmin, FAO, METI/NASA, USGS.**





## 3.2. Weather situations

Weather situations leading to HPE in the Mueglitz area are typically related to two different prevailing weather regimes.
Firstly, cyclones on the so-called Vb track defined by Van Bebber (Köppen, 1881, Messmer et al, 2015), which mainly occur in late spring or early summer. During a Vb situation, a low-pressure system propagates northwards from the Mediterranean over the Adriatic Sea and mainly influences the eastern part of Europe. It transports moist air from the Mediterranean Sea to the north, leading to heavy rainfall over large parts of Eastern Europe, especially over mountainous regions. Severe floods such as at the Elbe in 2002 and 2013 or the Oder in 1997 were caused by Vb cyclones as well as the above-mentioned flood
disasters at the Mueglitz where 2002, intensified by embedded convection, the precipitation rate reached 300 mm within one day.

A second mechanism causing hydrological extremes is heavy rainfall associated with convective systems. Convective rain often occurs as a result of isolated deep convective cells of limited areal extent or as organized cell systems along a frontal zone or convergence line. Due to their local character, these events can cause flooding of tributaries rather than flooding on
the larger scale of river basins.

During the 2019 campaign no severe floods were observed neither at the Elbe nor at the Mueglitz and its tributaries. On the contrary, the three summer months in the year 2019 were all drier than the multi-year summer averages. Especially in June and July, Saxony received only two thirds of the precipitation compared to the international reference period of 1961-1990 (DWD, 2020). A total of 6 IOPs (Table 3) were conducted during various convective events. The runoff coefficients of all
IOPs at the outlet of the Mueglitz catchment were very small, with a maximum of 0.05 for IOP4. Nevertheless, the IOPs allowed real world testing of the new instruments deployed, interdisciplinary cooperation, near real time (NRT) data exchange, as well as optimization of campaign conduction procedures and logistics. Due to the test character of the campaign, not all measuring devices were available continuously or at least during all IOPs, therefore their benefits were demonstrated along three examples within the period from 10 June to 16 July, with a specific focus on the IOPs 4 to 6.

**Table 3: Overview of IOPs during the MOSES 2019 Mueglitz campaign. Event types are stationary low-pressure systems with stratiform precipitation (SP), organized large scale convective systems (LCS), local scale isolated deep convection (IDC). Precipitation is averaged over the entire Mueglitz catchment (210 km$^2$). P indicates the total over the IOP, Pimax the maximum intensity detected during the IOP. Qmax is the maximum 15 min average flow rate observed during the IOP at the Mueglitz gauging station Dohna.**

| IOP | Start | End | Event types | P, $Pi_{max}$ | $Q_{max}$ (Dohna) | Runoff coefficient |
|-----|-------|-----|-------------|---------------|-------------------|--------------------|
| 1 | 19 May 17:00 | 22 May 20:00 | LCS + SP | 13.0 mm<br>4.2 mm h$^{-1}$ | 1.49 m$^3$ s$^{-1}$ | 0.02 |
| 2 | 27 May 18:00 | 28 May 21:00 | IDC + SP | 9.9 mm<br>1.3 mm h$^{-1}$ | 1.63 m$^3$ s$^{-1}$ | 0.01 |





| 3.1 | 04 June 07:00 | 05 June 00:00 | IDC | 0.6 mm<br>0.2 mm h$^{-1}$ | 0.89 m$^3$ s$^{-1}$ | 0.00 |
|---|---|---|---|---|---|---|
| 3.2 | 06 June 00:00 | 07 June 17:00 | IDC + LCS | 10.3 mm<br>9.5 mm h$^{-1}$ | 1.00 m$^3$ s$^{-1}$ | 0.01 |
| 4 | 10 June 20:00 | 13 June 12:00 | LCS | 33.3 mm<br>25.9 mm h$^{-1}$ | 5.13 m$^3$ s$^{-1}$ | 0.05 |
| 5 | 19 June 08:00 | 21 June 10:00 | IDC + LCS | 12.3 mm<br>15.0 mm h$^{-1}$ | 1.00 m$^3$ s$^{-1}$ | 0.00 |
| 6 | 11 July 08:00 | 15 July 18:00 | SP + LCS | 18.7 mm<br>7.1 mm h$^{-1}$ | 0.61 m$^3$ s$^{-1}$ | 0.01 |

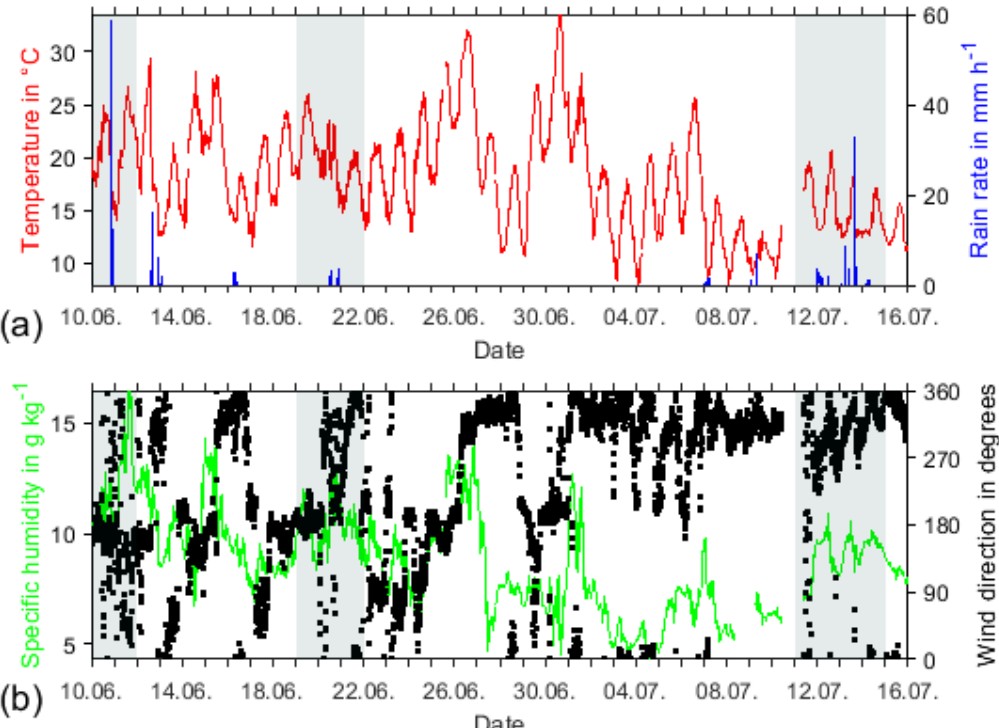


**Figure 4: Near surface measurements of meteorological variables at the main site from IOP4 to IOP6. Gray areas indicate the IOPs. The upper panel (a) shows air temperature (solid red line) and rain rate (solid blue line), the lower panel (b) gives the course of specific humidity (solid green line) and wind direction (black markers) in the given time period.**

The weather in spring and early summer 2019 was characterized by dry and warm conditions. In Saxony, June was 5.3 K (20.9

°C) and July 1.8 K (19 °C) warmer than the long-term average of the 1961-1990 reference period. At the same time, June with



44.3 mm (58.0%) and July with 46.7 mm (67.8%) saw significantly less precipitation than the reference period. Embedded in this warm and dry period (DWD, 2020) convective events occurred, triggered and marked the selected IOPs.

**IOP4**

IOP4 started on 10 June with the inflow of warm and moist air into eastern Germany, indicated by high values of specific humidity (approx. 12 g kg$^{-1}$) in combination with southerly winds (Fig. 4). Temperatures rose to values close to 30 °C (Fig. 4). A shallow low-pressure system formed in an unstable stratified subtropical airmass, triggering showers and thunderstorms in the afternoon. A line of thunderstorms reached the Mueglitz catchment at about 20:00 UTC and moved slowly further east. Zinnwald reported 35.6 mm of precipitation between 21:00 and 22:00 UTC. After 22:00 UTC the thunderstorm activity declined, however, light to moderate rain continued until 01:00 UTC. In total, between 20 and 50 mm of rain fell in the entire catchment area of the Mueglitz on this day. The main site received 31.8 mm (Fig. 4).

On 11 June the investigation area remained in a warm and moist airmass, but the main thunderstorm activity occurred over north-western Saxony. Only in the estuary of the Mueglitz some rain of 5 to 15 mm was observed.

Another line of thunderstorms (convergence line) quickly crossed the Mueglitz area between 15:30 and 16:00 UTC the next day (12 June). The precipitation amounts remained moderate with 10 to 20 mm along the Mueglitz. While the main site registered only 4.6 mm rain, the site of the German Weather Service at Dippoldiswalde-Reinberg, just a few kilometers to the northwest of Glashuette, reported 31.8 mm.

Due to the dry weather conditions in the preceding weeks, low flow conditions prevailed in the Mueglitz river at the onset of IOP4, characterized by a discharge of 0.61 m³ s$^{-1}$ at the gauging station Dohna. The maximum discharge of 5.13 m³ s$^{-1}$ was reached on 11 June at about 04:00 UTC, i.e., about 6 hours after the maximum thunderstorm activity in the catchment area. The second major line of thunderstorms on 12 June caused a secondary discharge peak of 3.87 m³ s$^{-1}$ at 20:00 UTC on 12 June.

**IOP5**

On 19 June subtropical air masses again made themselves apparent in the course of the day in a few isolated thunderstorms over Saxony. The subtropical character of the air mass is again indicated by a high specific humidity (>10 g kg$^{-1}$) and a southerly wind direction (Fig. 4). The Mueglitz area remained free of thunderstorms until the next morning. 20 June started as a warm and humid day (Minimum temperature 17.9 °C and 10 g kg$^{-1}$, Fig. 4). First rain showers occurred early. Between 05:30 and 08:00 UTC light and scattered precipitation was observed in the investigation area, hardly exceeding more than 1 mm. During the day, thunderstorm activity increased rapidly and the first strong thunderstorm formed on the Czech side south of the main ridge of the Ore Mountains around 11:00 UTC. This thunderstorm complex intensified rapidly and moved in a northeasterly direction. It reached the headwaters and the upper Mueglitz at about 12:00 UTC. At its edge, the complex triggered new thunderstorms and a line of thunderstorms crossed the entire investigation area until 13:00 UTC. After a mostly





dry afternoon, further showers and thunderstorms - albeit disorganized and not particularly strong - moved into the Mueglitz area or developed locally. They moved rapidly eastwards, adding another few mm of rain between 19:00 and 23:00 UTC. No significant precipitation was recorded after midnight and during the following day.

Overall, the maximum precipitation amount in the Mueglitz catchment during IOP5 was reached at the Zinnwald site with 405 22.5 mm, whereas for the largest part of the catchment less than 10 mm were observed (Fig. 7). The main site received only 3.3 mm (Fig. 4).

At the onset of IOP5, the discharge at the Mueglitz catchment outlet (gauging station Dohna) again fell back to baseflow conditions similar to those before IOP4, i.e., a discharge of 0.61 $m^3\ s^{-1}$. The overall low amounts of precipitation during IOP5 caused only a minor increase in river discharge, with a plateau value of 1 $m^3\ s^{-1}$ on 21 June from about 04:00 to 07:00 UTC.

**IOP6**

During the night of 11/12 July, a warm front brought warm and humid air to Saxony (Fig. 4). At 22:00 UTC, light to moderate rain began to fall in western Saxony. As it spread eastwards, the rain covered the entire catchment of the Mueglitz. The precipitation continued until 06:00 UTC with total amounts between 5 and 15 mm (main site 7.6 mm, Fig. 4), and around 20 mm near the Elbe. On 12 July the precipitation line moved eastwards with the warm front, while at the same time a secondary 415 trough aloft increased the precipitation activity. Convergence lines formed in the humid, moderately warm air, along which heavy showers and thunderstorms developed.

However, the precipitation activity remained low in the Mueglitz area where scattered showers formed between 10:00 and 12:00 UTC. Between 17:30 and 18:00 UTC the lower reaches of the Mueglitz received some rain and also some rain was observed near the Ore Mountains ridge between 00:00 and 01:00 UTC on 13 July. Between 04:30 and 06:00 a few rain showers 420 moved across the Mueglitz area from the north bringing light rain.

A northwesterly flow with moist and unstable stratified maritime air prevailed during 13 July. Further rain showers crossed the Mueglitz area between 08:30 and 10:00 UTC with very light rain. Coming from northerly directions, a more extensive rain area with embedded thunderstorms reached the Mueglitz catchment at 15:00 UTC. The heaviest rain occurred around 16:00 UTC, followed by two more hours of rain with decreasing intensity until 18:00 UTC. No more significant precipitation was 425 recorded during the night and until the end of IOP6 in the morning of 14 July.

Between the morning of 13 and the morning of 14 July 2019, rainfall occurred across the investigation area, ranging from 5 to 15 mm in 24 hours, with only small areas receiving around 20 mm (station network) and up to 40 mm (radar observations).



Due to the warm conditions without precipitation since the end of IOP5, the hydrological state of the Mueglitz catchment further became drier, characterized by a discharge of 0.22 m³ s⁻¹ at Dohna at the onset of IOP6. This discharge is close to the
long-term mean minimum low flow (MNQ) at this gauging station. The precipitation activities during IOP6 caused only a minor increase in catchment runoff, with a maximum discharge of 0.61 m³ s⁻¹ on 14 July 08:00 UTC.

### 3.3.    Atmospheric Moisture

Atmospheric moisture including clouds and precipitation is the main contributor to the water cycle locally. During the campaign we operated radiosondes and a profiling radiometer to measure the temporal evolution of the local vertical water vapor distribution in the atmosphere. However, moist air masses originate typically from far range advection of water vapor and clouds into the observational area. Therefore, it is necessary to combine the local observations with a large-scale meteorological model to understand the origin of advected air masses and the contribution of different source regions. In this
section, we describe such an approach by comparing first the model to the local measurements in order to approve its correct representation of the observed meteorological situation. Secondly, the model can then be used as a diagnostic tool to examine the origin of advected moisture.

The temporal evolution of water vapor anomaly relative to the mean water vapor profile measured by irregular balloon soundings is shown in Fig. 5 (left panel). The reddish colors between 19 June 2019 09:00 UTC and 20 June 2019 12:00 UTC
reveal rather dry air masses (up to -30% dryer than the mean profile) up to an altitude of 2 km and partly up to 5 km. Only above 5 km and up to the tropopause, air masses are moister up to 40%. With the passage of the thunderstorm front at ground (20 June 2019 12:00 UTC) moister air masses, up to an altitude of ~5 km are advected into the observation area (up to 50% moister than the mean). In the upper troposphere above 5 km aforementioned moist air masses are first replaced by very dry air masses (-50 to -60% dryer than the mean profile) around 20 June 2019 12:00 UTC and followed again by moist air masses
with enhancements of up to 50%. This moist signature reaches altitudes of up to 11.5 km at 20 June 2019 around 21:00 UTC and this varies close to the lapse rate tropopause at around 12 km. Additionally, the integrated water vapor (IWV) time series reveals the same variability and constitutes a good proxy to check whether the model is representative for the moisture distribution in the atmosphere. While the radiosondes provide atmospheric profiles at specific times (dashed vertical lines in Fig. 5) during the IOP, the radiometer enables a continuous observation of the IWV. Small data gaps visible during times with
precipitation and shortly thereafter are the result of raindrops remaining on the instrument's radome and disturbing the measurement, which is normal behavior for this type of instrument. The agreement between radiosondes and radiometer during times where radiosoundings are available is really good, which underlines the consistency of the in situ and passive remote sensing measurements. Even the agreement between ECMWF ERA5 and the observation is remarkably good although some periods with larger differences (19 June 2019 16:00 UTC until 21:00 UTC, and 20 June 2019 02:00 UTC until 04:00 UTC)
are discernible. Nevertheless, the general increase in IWV which is connected to the thunderstorm passage is well represented



in the data. In addition, cloud signatures within the ERA5 model (contour lines in Fig. 5) coincide directly with the observed moist anomalies, which further confirms the good representation of the meteorological situation in ERA5 and allow the usage of model data for further interpretation. The rather good agreement of cloud signatures between radiometer and ERA5 can also be seen by the liquid water path (LWP). In time intervals where the radiometer measure clouds (i.e. LWP above IWV) the model also simulates clouds.

Backward trajectories based on ERA5 are calculated 72 hours back in time starting every 30 minutes and every 100 m from the ground to 5000 m altitude at the Mueglitz catchment on a 0.05° x 0.05° latitude/longitude grid to investigate the moist air mass origin. The trajectories contain information about the horizontal and vertical advection and reveal regions with enhanced moisture uptake or release in the ERA5 model. For further analysis only those trajectories which reveal a moisture anomaly with relative humidities larger than 80% during the respective IOP were selected. During IOP5 humid air masses originated directly from the Atlantic Ocean with IWV values of more than 35 mm and were advected from south-west into the Mueglitz catchment as has been depicted in Fig. 6a. The mean cumulative water uptake along the airmass trajectories reveal overall drying with IWV values of up to -1 mm with some exception west of Portugal and north of the Pyrenees where air masses take up IWV with values of +0.5 mm. It has to be noted that the cumulative and local moisture uptake along trajectories exhibits smaller values than expected just from the difference between IWV of 35 mm over the Atlantic Ocean and observed IWV of 25-30 mm in the Mueglitz catchment during IOP5. This is mainly due to the divergence of air masses during advection and one would come to the expected values if all values of the entire map in Fig. 6 were summed up. The local moisture uptake/release within 30 minutes time intervals over central Europe during IOP5 can be seen in Fig. 6c. It is consistent with the cumulative IWV change that the maximum IWV release (-0.12 mm) is higher than the uptake (+0.08 mm) per 0.25° x 0.25° grid box. Still significant moisture uptake can be found over central France and southern Germany, while a strong IWV release signature connected with the precipitation observed during IOP5 can be found south-west of the Mueglitz catchment. The situation during IOP6 is more complex than during IOP5 (see Fig. 6b). While the moistest air masses can still be found over the Atlantic Ocean and reveal considerable cumulative drying of up to -1 mm during the easterly transport to the Mueglitz catchment, persistent moisture uptake can be found within air masses advected over central Great Britain, the North Sea to north and central Germany, whereas air masses advected from the central North Sea show general drying. The local moisture uptake in Fig. 6d exhibits an alternating pattern of moisture uptake and release regions over central Germany with maximum values of +/- 0.2 mm per 30 minutes and grid cell. This pattern can be attributed to precipitation and subsequent evapotranspiration of moisture due to two consecutive main precipitation events during IOP6 on 11-12 July 23:00-06:00 UTC and on 13 July 15:00-18:00 UTC. In general, the larger local moisture uptake values in comparison to IOP5 are in agreement with the higher amount of precipitation observed during IOP6 (Fig. 4 and Fig. 7).

This exemplary combination of different atmospheric observation techniques with an atmospheric model reveals their advantages for a good process understanding and interpretation of the measurements. The synergy between observations and model is necessary for all kinds of processes which either cannot be measured in the entire observational region or occur far





away but are still relevant for the general water budget. Important to note is that a model alone would not be sufficient because the observations are necessary to approve the model performance and reliability.

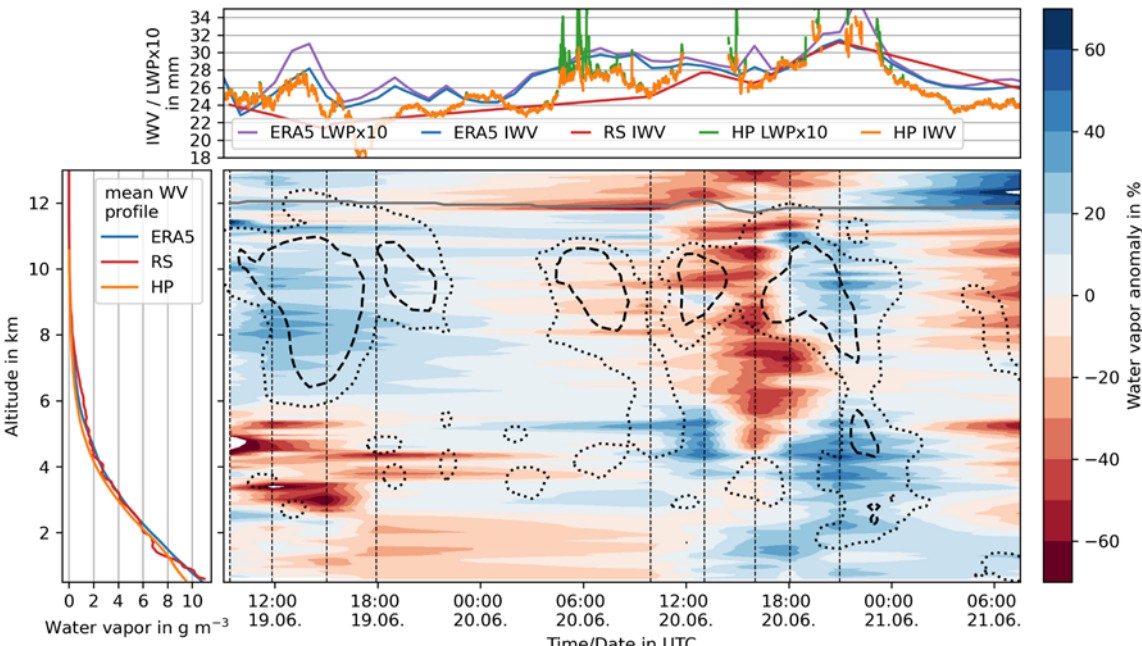

**Figure 5: Time series of measured water vapor anomaly in percent from mean water vapor profiles of radiosonde (RS), microwave radiometer (HP), and model (ERA5) (left panel) against altitude amsl during IOP5. Dashed vertical lines show performed balloon soundings, the grey line marks the first lapse rate tropopause calculated from the balloon temperature measurements. Dotted, dashed, isolines mark the cloud coverage (10, 50, and 100%) based on ECMWF ERA5 data, respectively. The upper panel shows the temporal evolution of the integrated water vapor (IWV) column. Red and orange lines show the balloon and radiometer
observations while the blue line represents IWV from interpolated ECMWF ERA5 data. The green and purple lines show liquid water path (LWP) multiplied by a factor of 10, added to the IWV for visibility reasons and represent clouds from the radiometer and model, respectively.**



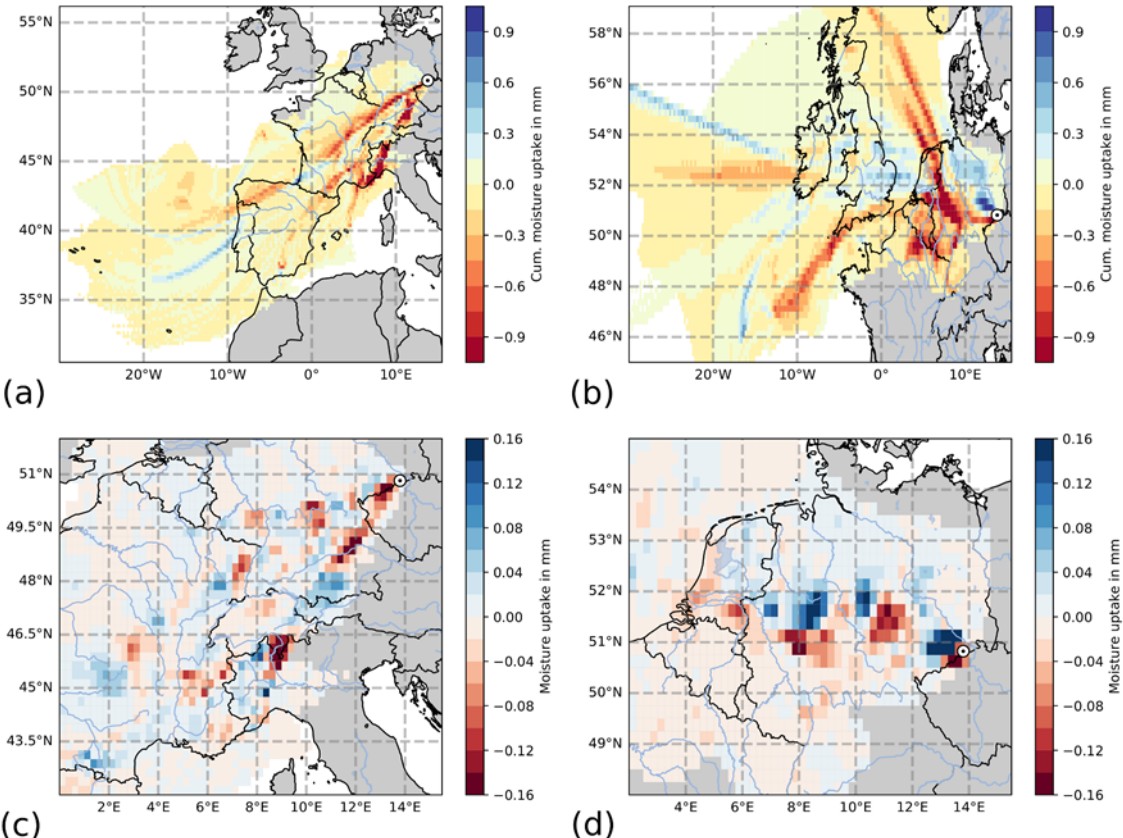

**Figure 6: Moisture uptake of air masses with relative humidity larger than 80% between ground and 5 km altitude in the observation region during IOP5 between 19 June 08:00 UTC and 21 June 08:00 UTC (a, c) and IOP6 between 11 July 08:00 UTC and 15 July 18:00 UTC calculated using air mass trajectories. The two upper panels (a, b) show the cumulative moisture uptake (blue) / release (red) of air masses during transport over central Europe. Lower panels (c, d) show the local moisture uptake/release mainly over Germany during the last approx. 12 hours before reaching the observation region. The white/black dot depicts the main measurement site in the Mueglitz catchment.**

## 3.4. Precipitation - Soil Moisture - Runoff

The analysis of HPE can be used to improve our knowledge of the links between precipitation and hydrological responses, in particular runoff generation. Thus, calibrated event precipitation data with high resolution in time and space for an entire catchment can provide significant insights into the small-scale variability of soil moisture and runoff, and thus the processes that drive these patterns.


For that purpose, the X-Band radar was installed on top of two containers at the main site. It scanned a range up to 100 km at a repetition time of 5 minutes and a radial resolution of 300 m. Although the antenna beam width is 1.3°, measurements are
stored every degree in azimuth. Nine elevations, starting at 0.7° to 30° with increasing steps were scanned. Whereas radar measurements provide volume filling data which allow investigation of vertical extent of precipitation, the surface rain intensity is derived from the lowest undisturbed elevation of the radar at every location in the measuring area. Mountains in westerly and southerly directions lead in these directions to measurements at high altitudes, partly above the altitude of precipitation formation. This causes strong underestimation (Fig. 7, a and c). Within the Mueglitz catchment the only relevant
obstacle for radar measurements is the wind turbine in close vicinity to the radar. It leads to a blind sector in an easterly direction from the main site in Fig. 7.

During IOP5 the average amount of precipitation measured by the radar within the Mueglitz catchment came to 12.3 mm. Its spatial and temporal distribution proved to be very inhomogeneous (Fig. 7a). While in the southeast of the catchment sums of
more than 63 mm were reached, large areas in the northern ranges received only 1 or 2 mm (Fig. 7b). More than 50% of total precipitation was registered between 12:00 and 13:00 UTC on 20 June, and an additional 2 mm in the late evening of the same day between 19:00 and 23:00 UTC (Fig. 8). Strong precipitation gradients both at the larger scale across the catchment (at about 1 mm near the confluence of the Mueglitz with the Elbe River at Dresden-Heidenau and about 50 mm at the Ore Mountain ridge) and at finer spatial scales are recognizable (Fig. 7b).
The area-average precipitation during IOP6 was slightly higher with 18.7 mm (Tab. 3). However, it did not feature the strong spatial gradients as IOP5 but considerable fine scale inhomogeneities which, in absolute values, were stronger than during IOP5 (albeit less visible in Fig. 7 due to the logarithmic scaling). The strongest contributions to the total event volume were measured on 13 July from 15:00 to 18:00 UTC (7.8 mm), from 11 July 22:00 UTC to 12 July 06:00 UTC (3.2 mm), and on 13 July from 04:30 to 06:00 UTC (1.4 mm), so this event was less dominated by isolated thunderstorms than IOP5 (Fig. 7).






**Figure 7:** **Spatial patterns of total precipitation measured by the X-Band radar for IOP5 (a+b) and IOP6 (c+d) for the entire area that is covered by the radar with 100 km radius (a+c), and for the Mueglitz catchment (black outline) (b+d). In (a) and (c) 45 circles indicate the in situ-point measurements by rain gauges of the German weather service (DWD). The data gap in the eastern direction is caused by the tower of a wind turbine in close vicinity to the radar.**

Both IOPs are dominated by strongly heterogeneous precipitation patterns. The applied event chain concept focuses on the impact of this heterogeneity on land surface processes, states and fluxes such as evapotranspiration, soil moisture, ground water storage, and runoff.

The temporal dynamics of near surface soil moisture at the main site was observed at the energy balance station (EBSM, Loc_1) and in its vicinity at five locations by a distributed wireless sensor network (WSN, see Schrön et al., 2018(b); Lausch

et al., 2018) (Loc_2 - Loc_6). All locations were equipped with soil moisture probes at depths of 0.05, 0.10, 0.15, 0.30, 0.45, and 0.50 m to obtain vertical profiles. Soil permittivity and soil temperature were measured at each depth with two redundant and slightly displaced sensors at 10-minute intervals. The monitoring locations were chosen to cover different soil types as well as different slope inclinations to capture a range of near-surface conditions. The near-surface soil moisture monitoring setup was complemented by a stationary CRNS at the main site. Figure 8 shows the time series of soil moisture at different depths at EBSM and from the stationary CRNS during IOP5 and IOP6.

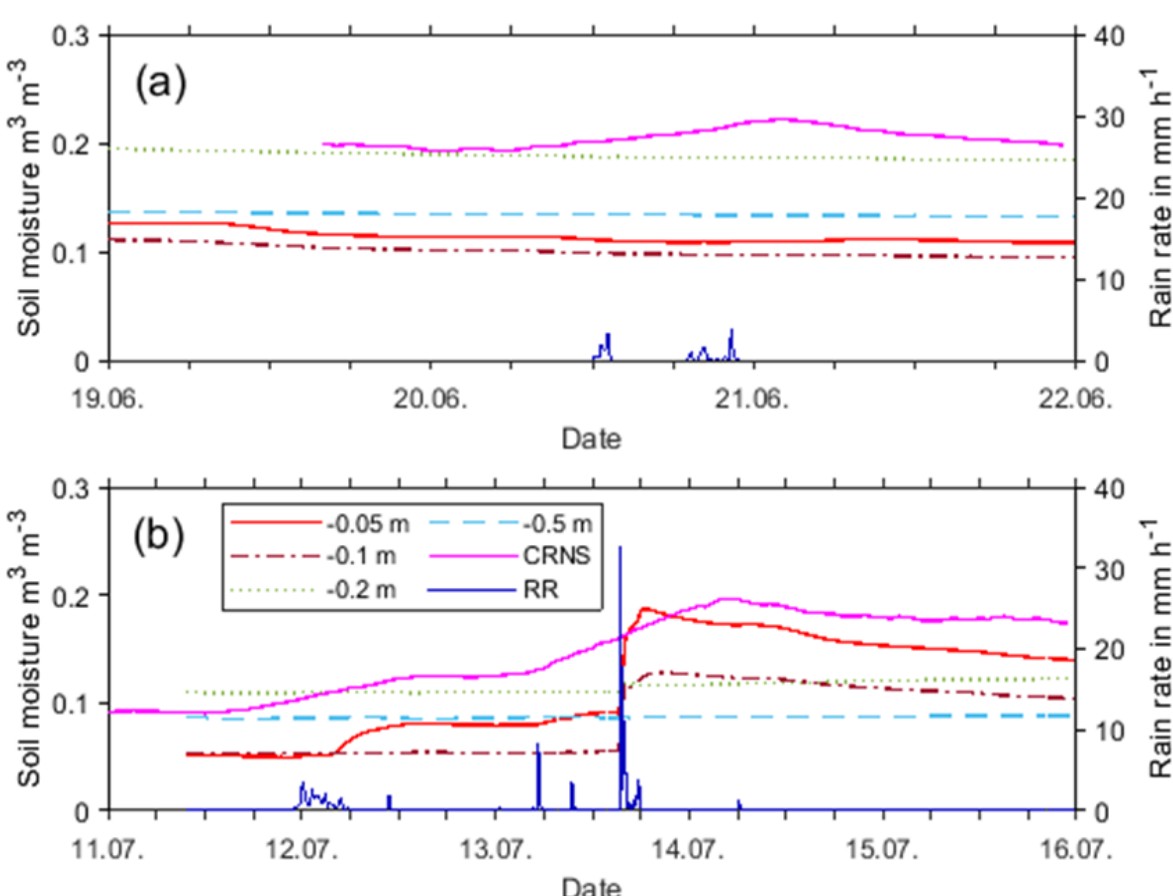

**Figure 8: Rain rate and soil moisture measured with probes at different depths as well as the integrative soil moisture of the upper 30 cm based on stationary CRNS at the main site, for IOP5 (a) and IOP6 (b).**

Figure 9 displays the soil moisture content data for all installed WSN sensors at all depths at Loc_1 - Loc_6. For comparison, the soil moisture values derived from the stationary CRNS measurements are shown at the top of the figure.





During IOP5, soil moisture decreased with time since the low total amount of precipitation (3 mm) that was distributed over a period of 9 h (Fig. 8) did not penetrate deeper than 5 cm into the soil and thus did not wet the soil layers where sensors were installed. The WSN sensors recorded an overall mean decrease of soil moisture of approximately 0.01 $m^3$ $m^{-3}$ at 0.05 m and 0.10 m, 0.009 $m^3$ $m^{-3}$ at 0.15 m, 0.0075 $m^3$ $m^{-3}$ at 0.30 m and 0.007 $m^3$ $m^{-3}$ at 0.45 m, 0.003 $m^3$ $m^{-3}$ at 0.50 m (Fig. 9). The CRNS data, in contrast, show an increase of wetness during the event, representing the interception of rainfall by the vegetation

cover and the soil surface (Fig. 8). While the CRNS is sensitive to water below and above the surface, the buried in situ sensors are representative only for a small integration volume around them.

All three of the WSN, EBSM, and CRNS were able to detect catchment drying during IOP5 and rewetting during IOP6. Warm and dry weather conditions between these IOPs (Fig. 4) resulted in an overall drying prior to IOP6, leading to a 30…50% decrease in soil moisture data at all depths (Fig. 8 and 9). Thus, soil moisture values after IOP5 varied from 0.11…0.21 $m^3$ $m^{-3}$

and these values decreased to a range of 0.03…0.10 $m^3$ $m^{-3}$ by the beginning of IOP6.

During IOP6, the higher precipitation compared to IOP5 had a more substantial impact on soil moisture (Fig. 8 and 9). The period with light rain (7.5 mm) in the early morning hours on 12 July led to an increase in soil moisture at a depth of 0.05 m. The major rain event (11.6 mm) on 13 July significantly increased soil moisture to a depth of 0.15 m. Deeper soil layers were hardly affected, and remained dry until the end of IOP6.

These observations at the main site may not be transferable to other sites, since radar measurements (Fig. 7) show strong heterogeneity of the precipitation patterns throughout the area. Hence, spatially distributed measurements are necessary to capture the soil moisture distribution in the catchment.



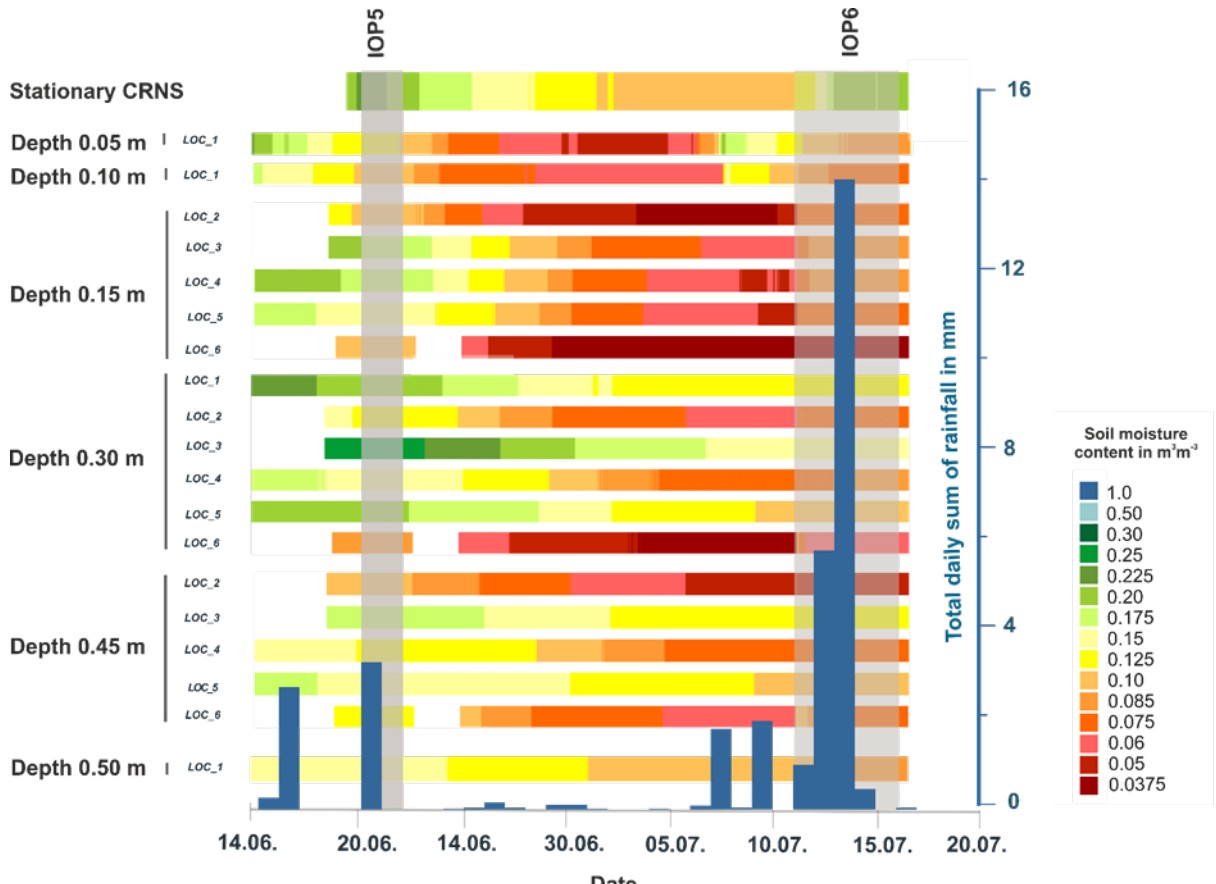

**Figure 9: Soil moisture of WSN sensors at all depths (0.05 - 0.50 cm) and monitoring locations (Loc_1 - Loc_6); integrative soil moisture of the stationary CRNS at the main site (top row); and daily rainfall measured at the main site.**

To this end, additional measurement campaigns were conducted with two portable CRNS units mounted on vehicles (UFZ Rover and GFZ Rover) to characterize the spatial variability of soil moisture in the Mueglitz catchment (CRNS roving). The WSN data served as reference points for the CRNS data processing, following Desilets et al. (2010) and Schrön et al. (2018(a)). Road and vegetation corrections were applied using OpenStreetMap road data and the CORINE land cover database 2019 provided by the Copernicus land monitoring service.

The CRNS roving method allowed us to quantify the effects of different land use types on soil moisture dynamics. We found that forests, followed by agriculture and urban areas, had the lowest drying rate (Fig. 10). This suggests that near-surface soil moisture in forests is more resilient to prolonged drying periods, while urban areas are at much higher risk of suffering from these events. The results from IOP6 further indicate that precipitation events literally evaporate on dry agricultural land under warm weather conditions, while forests are able to store precipitation water much longer in near-surface layers due to better radiative protection by the forest canopy, increased friction and therefore lower wind speeds, and adhesive soil structures. The





CRNS data also indicate a relation of the observed soil moisture changes with topography as also described in Guo et al. (2020)
and Garzón-Sánchez et al. (2021). For IOP6, a soil moisture increase was observed in the intermediate elevation zones of the
Mueglitz catchment (Fig. 10). This result may be explained by the fact that forests dominate in these intermediate, and often
steep, parts of the catchment. However, additional studies are needed to assess this result also in view of a slope correction of
the measured data and in combination with other remote-sensing and proxy data.

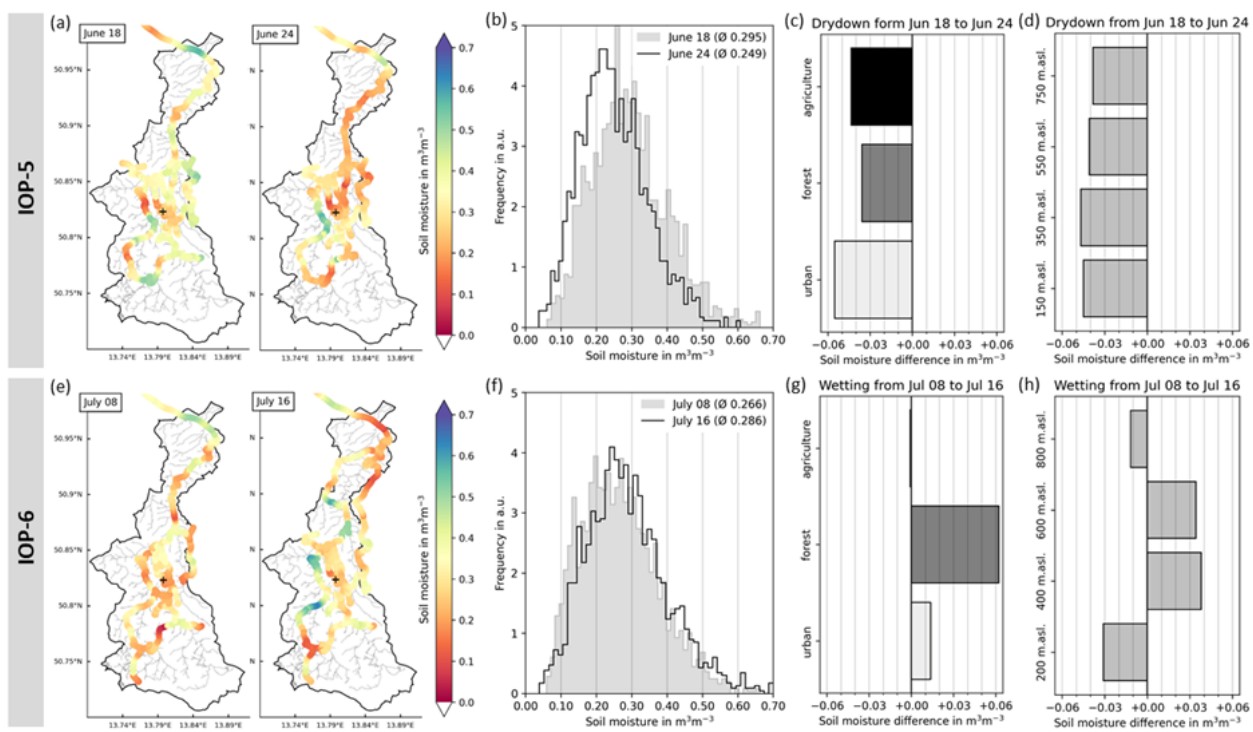


**Figure 10: Volumetric soil moisture measured with the CRNS rovers before and after IOP5 (top panel, a-d) and IOP6 (bottom panel, e-h) along the tracks in the Mueglitz catchment (see also Fig. 3). (a) and (e): Soil moisture along the route, interpolated to a 1 km resolution for illustrative purposes, the central cross marks the main site. (b) and (f): Normalized probability density function of CRNS measurements before (gray) and after (black) the IOP. (c) and (g): Mean change of soil moisture for agriculture, forest, and**
**urban areas as defined by the CORINE land use database. (e) and (h): Mean change of soil moisture for different elevation zones.**





**(a)**

**(b)**

**(c)**

**Figure 11: Runoff response of the Mueglitz river and of tributaries to precipitation events. (a) Time series of area-average rainfall intensities R and of discharge Q in the Mueglitz river and the tributary Ditterbach during IOP4. (b): runoff coefficients of the**
**Mueglitz catchment and the mean runoff coefficients of the 5 sub-basins for IOPs 4, 5 and 6. (c): Scatter plot of runoff coefficients of the Mueglitz catchment versus near-surface soil moisture (<0.15 m depth) measured at the main site, and maximum rain intensity for five events: IOP4 (10 June 20:45 UTC, 11 June 18:00 UTC, 12 June 16:00 UTC), IOP5 (20 June 11:15 UTC), and IOP6 (13 July 15:30).**





Another main target variable of the HPE monitoring concept presented here is river discharge. Discharge time series in the creeks surrounding the main site were based on water level observations at temporary gauges with LTC data loggers (Fig. 3 and 12). For the Mueglitz catchment as a whole, the discharge data of the official gauging station Dohna close to the catchment outlet were used.

The tributaries show a rapid response to rainfall (Fig.11a) for the example of the Ditterbach creek). The application of cross-
correlation analysis on the standardized time series of precipitation and river water level revealed that peak discharges in the tributaries occurred within less than one hour after the rain event. Close to the outlet of the Mueglitz catchment at gauge Dohna, event peak discharges are recorded with a lag time of 5 to 18 hours after the IOP rain events. In addition, the different sub-basin responses and lag times resulted in several consecutive peaks at the Mueglitz gauge for a single rainfall event.

Runoff coefficients were obtained by first calculating the total volume of direct runoff over the event time at each gauge, using
the constant baseflow separation method, and, second, by dividing it by the sum of event rainfall across the area of each sub-basin (Fig. 11b). In general, the resulting runoff coefficients of less than 0.02 in the tributary creeks were comparatively low (Dung et al., 2012). As described above, there were no heavy rain events during the IOPs, and catchment water storages in vegetation, soil and groundwater during the summer period were depleted, leading to minor runoff generation. Subsequently, the mean runoff coefficients of the five mostly forested sub-basins do not differ significantly among the IOPs. For the entire
Mueglitz catchment, slightly higher runoff coefficients occurred for some events. A dependence of the runoff coefficient on event-based near-surface soil moisture and rainfall intensity could be observed (Fig. 11c). This indicates that other parts of the catchment with different land cover, soil or precipitation characteristics contributed to a larger extent to its overall runoff response than the tributaries monitored in our campaigns. Thus, while the low-cost monitoring technique with LTC data loggers provided valuable runoff data of sub-basins including very small tributaries, a larger number of well selected
monitoring sites is still required to capture the large-scale picture of catchment runoff response.

### 3.5.    New Methods - Gravimetry

Given the relevance of catchment wetness conditions as one factor for flood generation, we suggest adding a novel component
to the HPE monitoring concept: terrestrial gravimetry. Terrestrial gravimetry is an emerging technology for non-invasive monitoring of water storage variations on the landscape scale of some hundreds to few thousands of meters around the instrument (e.g., Güntner et al., 2017). It is the only available technology for monitoring water storage changes in an integrative way over all relevant storage compartments, i.e., groundwater storage, unsaturated zone water storage, and, in some cases, snow water equivalents. Thus, it may even  enable measurements for the required ΔS in the water balance equation (Fig. 1).
The basic idea of gravimetry is the measurement of the acceleration of gravity at the Earth's surface which varies in space and time according to Newton's law of mass attraction as a function of the mass distribution and its variations above and below the terrain surface. The sensitivity of current generations of gravimeters enables the monitoring of mass variations that are due to water storage changes in the surroundings of the instrument, which are about 7 or 8 orders of magnitude smaller than the





gravitational attraction by the Earth's mass itself (e.g., Van Camp et al., 2017 for an overview). With absolute gravimeters,
gravity is derived from observing the trajectory of a free-moving object along the vertical. Relative gravimeters determine
changes in gravity by recording the related deviations of a test mass from its reference position, either continuously over time
or as gravity differences between different observation points.

For the HPE monitoring setup we suggest a hybrid approach of (1) continuous relative gravity monitoring at a reference station
within the study site (the main site in Fig. 3) to get a continuous time series of water storage changes, (2) time-lapse relative
gravity surveys at several network points throughout the study area to assess spatio-temporal variations of water storage before,
during and after the event, and (3) occasional absolute gravity measurements at the reference station to correct for the
instrumental drift of the relative gravimeter. Advancing the concept of hybrid gravimetry used in some previous studies
(Naujoks et al., 2008, Hector et al., 2015, Chaffaut et al., 2022), we adopt the following new components:

(i) For continuous gravity monitoring at the reference site we deploy a gPhoneX relative gravimeter instead of a
superconducting gravimeter (SG). The gPhoneX (Fig. 12b) is considerably smaller, lighter and with considerably less energy
consumption than a SG and thus more suitable for the required HPE approach with comparatively easy to deploy and short-
duration field installations. A gPhone is a relative gravimeter based on a zero-length spring system which measures gravity
changes in a temporal resolution of 1 second with a precision of 1 µGal, manufactured by Micro-g LaCoste, Inc.

(ii) For continuous operation of the gPhone reference station at any point of hydrological interest in a HPE study area and
independently from the presence of larger infrastructure such as a building and a connection to a power line, we developed an
energy self-sufficient container to house the gravimeter and auxiliary monitoring devices, the so-called gPhone Solar Cube
(Fig. 12a). This field container, built out of a square 6 feet sea container, protects the gravimeter from the natural environment
and ensures stable monitoring conditions in terms of air temperature and humidity. At the same time, the footprint of the
housing is kept small so that deviations in the gravity measurements due to the umbrella effect, i.e., the disturbance of the
natural hydrological conditions in the direct surroundings of the gravimeter (e.g., Reich et al., 2018), can be kept small. The
Solar Cube comprises a battery system for continuous power supply, an electronic power module for operating, monitoring,
logging and for online data transfer, eight solar panels mounted on the sides and on the rooftop of the container for power
input, as well as isolated walls and ceiling in order to reduce inside temperature variations. Within the container, the gPhoneX
is placed on top of an ODIN leveling platform that compensates minor tilts of the instrument that may affect the observed
gravity. The leveling platform in turn is placed on a stable pillar-table system made of steel, which is mounted to a small (about
0.5 m x 0.5 m) concrete foundation in the sub-surface at a depth of 0.5 to 1.0 m. The pillar is not connected to the container in
order to avoid container vibrations propagating to the instrument. Besides the gravimeter, standard instruments used at weather
stations are deployed at the outside of the container, including sensors for air temperature, relative humidity, air pressure,
horizontal wind speed and wind direction, net radiation, as well as a cosmic ray neutron probe for soil moisture monitoring.
(iii) Instead of absolute gravimeters of the type FG5 used in the previous setups of hybrid gravimetry, we suggest applying an
atom quantum gravimeter (AQG) for performing the repeated absolute gravity measurements to correct for the instrumental


drift of the gPhone. An instrument of the Muquans / iXblue AQG B-series (Cooke et al., 2021), specifically adjusted for outdoor operation, is used for this purpose.

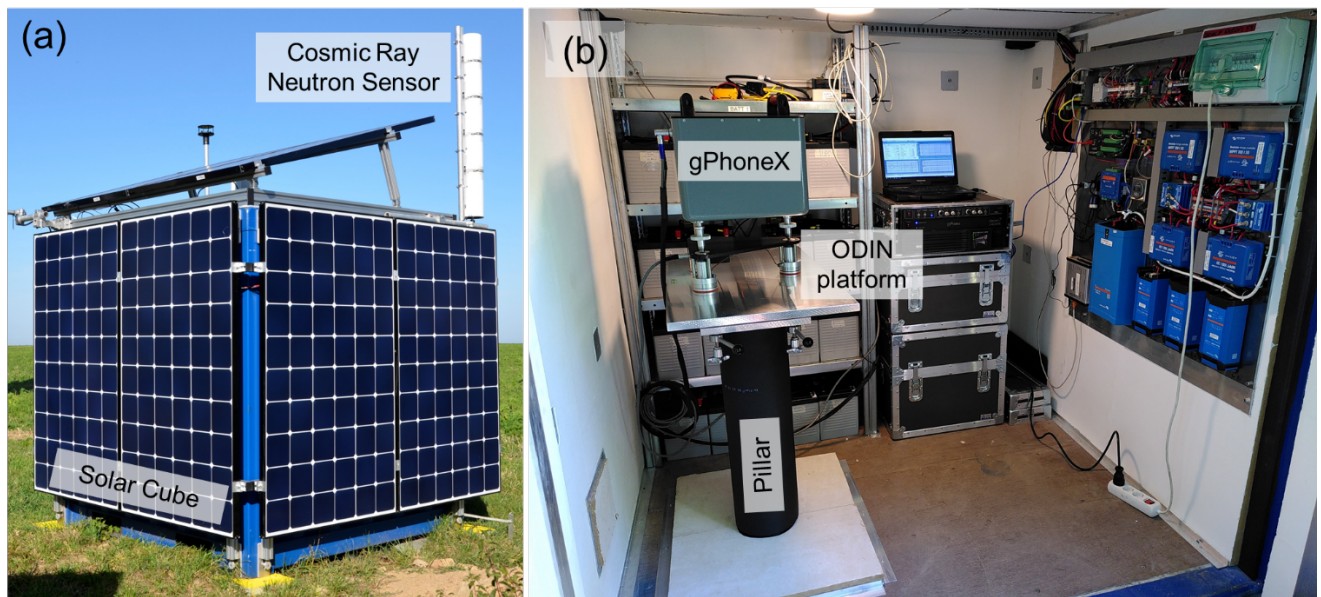


**Figure 12: The gPhone Solar Cube at the main site, (a) outside view with solar panels and hydro-meteorological sensors and (b) inside view with gravimeter, power units, and operating systems.**

For deriving spatial patterns of gravity changes and thus water storage changes during IOPs, gravimetric field surveys on a

network of monitoring points throughout the study area were conducted. The surveys were usually realized both at the beginning of the IOP just before the event to monitor pre-event conditions and right after the precipitation event ended. The surveys were carried out with two CG-6 relative field gravimeters (manufactured by Scintrex Ltd.) at nine network points (Fig. 3), using small concrete pillars of the official geodetic monitoring network of the Federal State. All surveys started and ended with measurements at the reference site to link the survey data to the continuous gravity time series of the gPhoneX. At all

network points, measurements were taken simultaneously with both devices on the same pillar for a time period of 10 minutes. In parallel, a CRNS probe, installed within the campaign vehicle, was operated to have an estimate of near-surface soil moisture for each gravimeter record. On completing both survey runs within one IOP, data processing aims at the differences in gravity values at each field site, resulting from pre- and post-event water storage conditions. Negative differences would thus indicate a reduction in water mass in the natural system whereas a positive difference is expected here at most sites because of the

accumulation of water mass during the precipitation event. The topographic position of the network point, however, may modify the gravity effect of water storage variations and needs to be individually considered at each point.

The gPhone Solar Cube was deployed late June 2019 and continuously operated throughout IOP6 and also the gravimetric network campaigns were carried out. Interpretable gravity data for hydrological applications, however, could not be achieved



primarily because the gPhoneX time series turned out to be very noisy and exhibited many major offsets which might be
attributed to the operation of the nearby wind turbine including its power transformation unit, both having adverse impact on
the gravimeters by vibrations and electromagnetic fields.

## 4.        Discussions

The MOSES measurement campaign in the Mueglitz catchment in 2019 was the first implementation of the new event-oriented
observation concept for short-term HPE, deploying the mobile observing systems to cover the entire process chain from
atmospheric transport processes and precipitation formation, over land surface and subsurface water flow and storage, to runoff
dynamics. Designed as a test campaign, it initiated the required cooperation of scientists from different disciplines
(meteorologists, hydrologists, geophysicists), to establish and optimize campaign logistics and deployment procedures, yet
illustrating the added value of such interdisciplinary cooperation. It has to be noted that the weather situation during the study
period was not favorable for dedicated HPE campaigns:  the overall rainfall amount in the summer season of 2019 was only
two thirds of the long-term mean summer precipitation of the study area; dry soil conditions prevailed at the onset and
throughout the period of investigation; low discharge values of about one third compared to the long-term mean annual
discharge prevailed at the Mueglitz gauge in Dohna; and the forecasted heavy precipitation events turned out to be of only
small to moderate intensity and volume. Nevertheless, valuable experiences on performing HPE campaigns could be collected
and insights into the hydro-meteorological functioning of the study area were gained.

Given the initial focus on short-term HPE and flood generation, for which evapotranspiration tends to be of minor relevance,
we did not put particular emphasis on comprehensive ET-monitoring in our observation concept. Nevertheless, by combining
ET observations of three sites with our considerable effort to determine precipitation and soil moisture with high temporal and
spatial resolution, and by including the runoff observations, we are able to make a tentative effort to assess the water balance
of the events (Fig. 1), both at the local scale of the focus site and at the regional scale of the entire Mueglitz catchment. This
is a first step towards a comprehensive and holistic understanding of the water flux and storage dynamics during HPE.

This is illustrated in Fig. 13 by example of an 11-day period around IOP6, starting on 08 July 00 UTC one day before the first
precipitation in the study area and ending on 19 July 00 UTC when discharge at the Dohna gauge fell back to the pre-event
baseflow value. For the catchment scale with an area of 210 km² (Fig. 13a), precipitation was determined from the radar
measurements, covering almost the entire catchment at 5-minute resolution. Evapotranspiration was assessed by daily ET
measured at three energy balance stations, one at the main site, and two operated by the Technical University Dresden
(Goldberg et al., 2008) in Oberbaerenburg (Fig. 3) and Tharandt (50.963 N; 13.565 E, not shown).  ET differences among
these three stations were on average 46% on the daily scale. This only shows once again the need for representative locations
of energy balance stations in the measurement area.

However, the average daily ET values of the three stations were used for the water balance analysis presented here. Soil
moisture was taken as an area-average from CRNS along the rover tracks depicted in Fig. 3, and as illustrated for the two





roving campaigns before and after IOP6 in Fig. 10. As the in situ soil moisture measurements by the buried sensors at the main site indicate that rainfall during IOP6 increased soil moisture at a depth of 0.1 m but not at depths of 0.2 m and below (Fig. 9), it was assumed that the absolute soil moisture change observed by CRNS concerns the top 0.15 m of the soil only. Thus, this

depth was used for converting the CRNS-based soil moisture changes from the percentage of volumetric water content to an actual storage change. For runoff, the discharge time series at the Dohna gauging station was used.

For the water balance analysis at the local scale (Fig. 13b), the measurements at the main site were used. Precipitation was taken from a PARSIVEL disdrometer with a detection surface of 50 mm$^2$. Evapotranspiration represents the fetch of at least 400 m upwind of the energy balance station at the main site, covered by a corn field during the IOP. Soil moisture was measured

by the stationary CRNS sensor with a footprint of about 10$^6$ m$^2$ and a depth of moisture change of 0.15 m, following the in situ sensor observations (see above). In absence of an operational gauge at a creek close to the main site during IOP6, the large-scale runoff given by the gauge data from Dohna was assumed to be applicable also at the local scale.

In line with the high rainfall heterogeneity observed throughout the study area (Fig. 7), precipitation at the main site deviated from the basin-average by roughly twice the volume during the IOP6 period (Fig. 13). Given the overall dry conditions in the

catchment, runoff was of minor importance for the overall hydrological budget during the campaign period. Little of the rainfall was converted into runoff; instead, it was attributed to storage or it evaporated. In fact, both on the local and regional scale, ET was the dominant component of the water balance during the warm and dry period considered here, with an average daily ET of 3.2 mm per day on the catchment scale and a slightly higher value of 3.8 mm per day at the main site. The rainfall input at the main site led to a corresponding increase of water storage in the uppermost soil layers as observed by CRNS (Fig. 13b).

Notably, on the regional scale, the catchment-average increase of soil moisture during the event period was considerably smaller, based on the two CRNS roving campaigns about three days before and three days after the event (Fig. 13a). Reasons can be (i) the overall smaller amount of rainfall at the catchment scale, (ii) the effect of three days of ET that already reduced water storage after the rainfall until the CRNS measurement was taken, (iii) the limited coverage of CRNS roving that may have missed sub-areas of the catchment with a higher soil moisture increase, (iv) an increase in water storage by preferential

infiltration processes at soil depths that are deeper than those captured by the integration depth of the CRNS method.





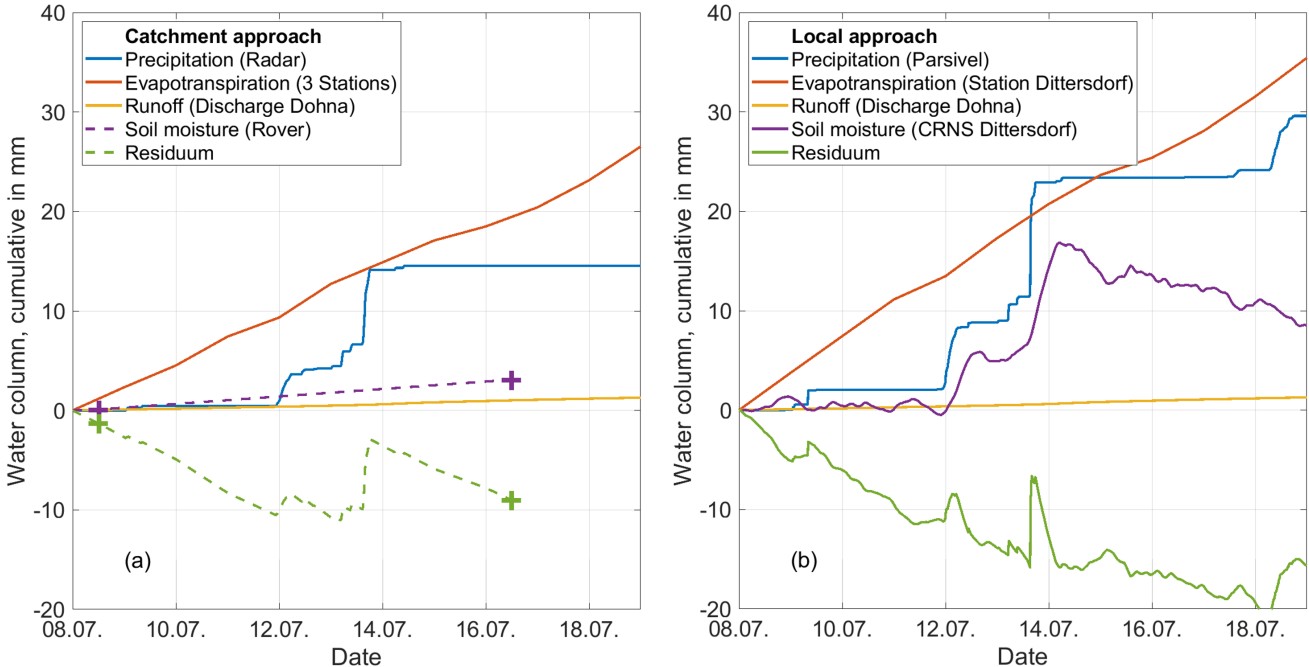

**Figure 13: Cumulative dynamics of water balance components around IOP6 at (a) the Mueglitz catchment scale using area-average precipitation from X-Band radar, averaged evapotranspiration from three EB stations (main site, Oberbaerenburg, and Tharandt), and averaged soil moisture from CRNS roving and (b) the local scale using point measurements at the main site for precipitation, evapotranspiration, and soil moisture.**

The residuals shown in Fig. 13 point to a marked imbalance of the water budget as assessed with the deployed measuring systems, both on the regional and local scales. In spite of uncertainties of the measuring systems that may partly explain these residuals, some general features can be identified. In particular, the comparatively large water loss by ET is not reflected by a corresponding decrease of (CRNS-based) near-surface soil moisture. This indicates a major contribution of the deeper unsaturated zone and of groundwater to the overall water storage change in the study area, in particular to accommodate the ET demand during this period. This assumption can be well justified given the widespread vegetation cover of the catchment, including forests with root zones that largely exceed the CRNS measurement depth, which is also the case for the corn surrounding the main site (Ordóñez et al., 2018). Therefore, our HPE monitoring concept additionally includes terrestrial gravimetry to fill this observation gap. This technique is sensitive to the entire unsaturated zone and to groundwater (see Subsection 3.5), and captures water storage changes that occur at larger depths than what could be achieved with soil moisture monitoring techniques. Albeit not providing data with sufficient quality during this test campaign, there is a prospect that gravimetry can provide relevant information to explain residuals such as those presented here (Fig. 13), and contribute to closing the water balance during HPE.



The observed differences in the dynamics of water flux and storage terms the local versus the regional scale underline the need to combine different observation techniques with their respective spatial and temporal measurement scales. For example, the CRNS-based roving campaigns, while with lower temporal resolution, revealed the different area-average soil moisture
dynamics of the catchment relative to local scale observations. Accordingly, the distributed survey concepts for CRNS-based soil moisture and gravity-based total water storage that rely on one or more reference stations with continuous observations and multiple survey points or tracks throughout the study area, covered by time-lapse measurements, are considered as essential components of the HPE monitoring design.

**5.        Summary and Outlook**

A new cross-disciplinary observation strategy for heavy precipitation events from rainfall formation to flood runoff generation was first applied at the MOSES field campaign in the Mueglitz catchment (eastern Ore Mountains, Germany) from May to July 2019. Meteorologists, hydrologists and geophysicists of four Helmholtz Institutions collaborated to observe the entire process chains from the source of atmospheric moisture to runoff dynamics in a river catchment. An event-oriented observation
concept has been set up, based on mobile and flexibly deployable measuring systems and a campaign design and operation concept that is optimized for convective HPE with rather short forecast lead times.

Three examples illustrate the advantages and new opportunities that this measuring concept provides towards a better understanding of HPE and flood processes, in combination with data from institutional measurement networks operated by national or local authorities as well as with modeling approaches. First, atmospheric moisture transport processes from the
source over the Atlantic, moisture uptake and loss when passing over landmass to the temporal evolution of integrated water vapor, liquid water content and vertical moisture distribution over the Mueglitz catchment can be traced and deciphered. Second, the link from precipitation measurements with high spatial resolution, over regional soil moisture patterns to the runoff dynamics of the Mueglitz and of its tributaries can be established. Third, a perspective to assess catchment total water storage variations during an event in a more holistic way that integrates over all storage compartments and thus provides the full ΔS
term in Fig. 1 is established by adding terrestrial gravimetry as a new component to the measuring concept. By closing the water budget in this way also on short time scales of an event, an important contribution to a full understanding of event dynamics can be made.

The cross-disciplinary design opened up new opportunities to address the different terms of the water budget in a catchment on both local and regional observation scales. Thus, in view of large spatial heterogeneity, it can be expected that the
contribution of small-scale dynamics to the overall catchment response can be unraveled in a better way by the observation approach presented here. Especially for convective driven HPE over complex terrain when highly heterogeneous precipitation falls on inhomogeneous soil types with different land use, it is essential to combine observation techniques with different spatial and temporal resolutions: X-Band radar measurements deliver high resolution precipitation data in time and space for individual (sub-)catchments, but require disdrometer or rain gauge networks for calibration. The estimation of





evapotranspiration requires several stations that cover the range of predominant soil types, land use and altitudes - a requirement that was not fulfilled within this campaign. The mobile CRNS soil moisture measurements along rover tracks give access to the urgently needed information about spatial soil moisture distribution and advanced catchment-average soil moisture values that can be expected to be superior to those based on individual point measurements. CRNS roving needs to be combined though with stationary, continuously measuring CRNS and soil moisture sensors networks to cover the temporal

dynamics and to assess the actual penetration depth. More frequent rover tracks are desirable in order to improve time resolution, but they are limited by their manpower requirements. Terrestrial hydro-gravimetry as an emerging technology for non-invasive and integrative monitoring of the water storage term also applies this hybrid concept of both continuous and spatially distributed monitoring with a reference station and time-lapse surveys with mobile relative gravimeters on a network of sites within the catchment.

The campaign operation concept that included an alert and decision making process for IOPs based on meteorological forecasts was well suited. These alerts led to IOPs starting early enough to allow for performing observations of the pre-event catchment conditions as one of the important factors for understanding the catchment response to a HPE. However, the approach adopted here for setting the IOP end to just a few hours after the end of the rain event needs to be revised in future. It may be preferable to set the IOP end to a later point in time, eventually to the moment where discharge has fallen back to its pre-event value, in

order to capture the runoff recession behavior of the catchment. This will support a more comprehensive assessment of the event water budget.

Because the extremely dry summer of 2019 did not provide extreme precipitation, the HPE observation concept could only be tested on less intense events. For future campaigns it is therefore planned to combine HPE deployment concepts with those developed for emerging heatwaves and droughts. This strategy not only increases the probability of encountering the desired

weather extremes, but also releases synergies in the use of equipment, personnel and scientific knowledge gain. As the follow up campaign scheduled for 2020 in the Mueglitz catchment fell victim to the Covid-19 pandemic, the insights gained with this first test campaign were incorporated into the Swabian MOSES campaign of 2021 (Glaser et al., 2022, Kunz et al., 2022).



**Data availability:**

The MOSES Data Discovery Portal is accessible at https://moses-data.gfz-potsdam.de/onestop/#/ and provides MOSES campaign metadata and datasets. All raw data measured by the MOSES RI can be provided by the corresponding authors upon request. Radiosonde data are available from http://www.tereno.net/geonetwork (Forschungszentrum Jülich IBG-3, 2022). ECMWF ERA5 data are available at https://cds.climate.copernicus.eu/cdsapp#!/home (Copernicus Climate Change Service, 2017). The runoff data for the Mueglitz River are available from the Saxon State Office for the Environment and Geology and can be downloaded here: https://www.umwelt.sachsen.de/umwelt/infosysteme/ida/. The levellogger data presented in subsection 3.4 are available in a data publication by Nixdorf et al. (2021). The rain gauge data can be downloaded from https://opendata.dwd.de/climate_environment/CDC/observations_germany/climate/1_minute/precipitation/historical/.

**Competing interests**:

The authors declare that they have no conflict of interest.

**Author contribution**:

All authors contributed to the conception, organization and design of the campaign and of the monitoring systems. Andreas Wieser was the responsible campaign coordinator. Material preparation, data collection and analysis were performed by Andreas Wieser, Andreas Güntner, Jan Handwerker, Dina Khordakova, Uta Ködel, Martin Kohler, Erik Nixdorf, Marvin Reich, Christian Rolf, Martin Schrön and Claudia Schütze. The draft of the manuscript was written by Andreas Wieser, Andreas Güntner, Christian Rolf and Claudia Schütze. All authors complemented and improved the manuscript. All authors read and approved the final manuscript.

**Acknowledgements:**

This work was supported by funding from the Helmholtz Association within the framework of MOSES. We acknowledge funding from the Initiative and Networking Fund of the Helmholtz Association through project "Digital Earth" (funding code ZT-0025). This project made use of the facilities that are part of the KITcube (Kalthoff et al., 2013) and TERENO (Zacharias et al. 2011). This work was partly funded by the DFG (German Research Foundation) via the research unit FOR 2694 Cosmic Sense (grant no. 357874777).-We thank all participating Helmholtz centers for their additional personal and financial support. The execution of the measurement campaign would not have been possible without the tireless efforts of the technical staff, even on weekends and at night. We thank the Chair of Meteorology of the TU Dresden for providing us with the evapotranspiration data from the anchor station Tharandt and the station Oberbaerenburg. The European Centre for Medium-Range Weather Forecasts (ECMWF) is acknowledged for ERA5 meteorological data support, and the German Weather Service for network data and support at the Zinnwald site. Our special thanks go to Raik Bellmann of the Liebenauer Agrar GmbH, and the landlords Susanne and Jens Höhnel for their support at the main site, councilor Bernd Grahl as well as the local population for their support and interest. Finally, we also thank Paul Ronning for proofreading the manuscript.



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
