# Peer review of "First implementation of a new cross-disciplinary observation strategy for heavy precipitation events from formation to flooding"

_Hydrology and Earth System Sciences, 2022_

## Author Comment (AC1)

**Reply on RC1:**

We cordially thank the anonymous reviewer for the valuable comments and the recommendations for the improvement of our manuscript.

In terms of the mentioned two minor comments of reviewer 1 *(in italic)* we are going to revise the manuscript:

*1)        The paper is too long for me. I believe the authors may consider the reduction of some parts, avoiding to go too much in details in the description of the technological aspects of the measurement apparatus. Likely these descriptions can be moved in the supplementary material.*

**Reply:** In accordance with the reviewer's suggestion and also taking into account the comments of reviewer 2, we will substantially shorten the main manuscript in the revised version. We will move most of the detailed technical descriptions of the monitoring devices, including the gravimetry approach, and the detailed description of the weather situation during the intensive operation phases into the supplementary material. By this way, we expect to significantly improve the readability of the paper, and to focus on its main messages.

*2)        The HYDRATE project (https://cordis.europa.eu/project/id/37024), ended in 2010, has been specifically designed to monitor flash floods and hence with a purpose very close to that addressed by the authors in this study, and in the future monitoring campaigns. I would suggest to make a link to this project also to benefit from the experience gained.*

**Reply:** This is indeed a relevant study, thank you for mentioning it. It will be carefully considered and reference to it will be given in the revised version.

---

## Author Comment (AC2)

**Reply on RC2:**

The authors thank Dr. Rolf Hut for the valuable comments and his thoughtful review including recommendations to improve our manuscript. Based on his comments and the comments of reviewer 1, we are going to prepare a revised and shortened manuscript that (among other changes) we expect to be more focused on the main topics of the study.

We would like to address the comments of Rolf Hut *(in italic)* more in detail:

*1)      The authors present the results of an extensive field work campaign intended to capture data on heavy precipitation events (HPE). The document describes many aspects of this campaign and this is directly my top concern with the document: it lacks focus. I have identified three major topics in the manuscript which are all independently more than worthy enough of their own manuscript, but together create a manuscript where it is unclear for the readership what they can take away from reading it.*

*The **three topics I have identified** are (and let it be said that maybe the authors would have come to a different list)*

1.  *'an elaborated event-triggered observation concept' (line 46)*
2.  *'closing the catchment water balance at the HPE scale' (line 58)*
3.  *'new measurement system based on terrestrial gravimetry' (line 106)*

*I would suggest to chose one of these topics and focus the paper on that and write two seperate (short) papers on the other two topics. This gives the readership a better choice in what to read (amids the never ending mountain of literature to read and keep up with). I fully understand this is not something the authors are happy to hear given the amount of work involved, but I trully believe it will lead to a better collection of papers. If the choice is made to stick with all of this information in one paper, I would strongly suggest to organize the paper along the lines of the three topics mentioned above. I will leave, as standard, the decision on how to proceed with the editor.*

**Reply:** The authors agree with the reviewer that the paper covers several topics as those listed above. However, we consider them to be closely linked to each other and to be integral components of the HPE monitoring setup that we want to introduce here. Therefore, we think that we cannot separate out these topics into three separate papers as we would like to present the MOSES approach comprehensively as a flexible monitoring observation system and the campaign in the Mueglitz catchment as an example of the implementation of the research infrastructure. Nevertheless, we agree with the reviewer that we need to re-organize the paper in terms of clarification of these main topics and length (as also recommended by RC1). We will thus move larger parts of the detailed device descriptions including the gravimetry approach and the detailed description of the weather situation during the intensive operation phases into the supplementary material. This revision will refine the paper in terms of making the essential messages, especially the comprehensive cross-disciplinary and event-triggered approach more prominent and will ensure significantly improved readability of the paper.

*2)      C**omment on 'an elaborated event-triggered observation concept'***

*I really like the idea of the multi tier observations where based on forecasts the team switches to 'intensive monitoring'. It would really help the readership if the 'pre-defined environmental parameters' (line 144) used to make that decision was shared and even more: if the design process to come up with these parameters was shared for others to use in their own campaigns. I believe that this is a vital part of the innovation of this part of the manuscript and should be elaborated on. Related to this is that the*

*colors in table 1 need explenation: when and how is the final decision for an IOP made based on the inputs?*

*The observation methods chosen cover a wide range of athmospheric and hydrologic interesting parameters that relate to HPE and the water balance in general. However, it is unclear what the selection of instruments is based on. Line 170 states that "To determine the water balance components according to Fig. 1 microwave radiometers, Doppler lidar and ardio soundings provide information on the state of the atmosphere as well as changes in water vapo distribution..." Fig. 1 contains many more processes, so the authors need to present a justification on how the instruments used in the campaign were chosen. And honestly: "we had these amazing things and wanted to use them" would be justification enough in my book, but that has to be acknowledged.*

**Reply:** This is a very good point which we will take up during the revision of our manuscript. For that purpose, we are going to sharpen the text from line 160 onwards, also to make the content of our decision matrix in Tab. 1 clearer to the audience. In that way, we will reconsider the colour scheme of in Table 1 and give and explain the set-up procedures for the threshold values of the parameters taken into account. Given that the detailed technical description of the applied monitoring techniques will be moved into the appendix, few paragraphs will remain in the main text of the revised manuscript that we will adjust according to the reviewer's suggestion to make it more clear which observation technique has been selected for which reason as a component of the overall structure of the HPE monitoring approach.

**3) Comment on 'closing the catchment water balance at the HPE scale'**

*I first and foremost want to stress that I 'm strongly against 'closing the water balance' as a goal in and of itself. Given the point based nature of most of our measurements, as well as the impossibility to measure a lot of the processes that transport water into and out of catchments, I don't see how we can ever 'close' the balance. However, as a thinking concept it can still serve its purpose. Having said that, I was surprised to see ERA5 used as part of the precipitable water calculation. If a re-analyses data source, based on both model and observations, is used in this part, why not also in determining rainfall? Or ET? I think it makes the analyses stronger if the authors either totally rely on their own observations, or use all the available (satellite / re-analyses) data in all aspects of the water balance study.*

**Reply:** We agree with the reviewer that closing the water balance is a huge challenge. For a long time, the unavailability and/or high uncertainty of observation techniques has hampered or even prevented efforts in going into this direction. Nevertheless, there is now an increasing number of observations methods around that go beyond the point scale, and there is one option to monitor for the first time at all the so far unobservable component of the water balance besides precipitation, evapotranspiration, and runoff, i.e., gravimetry to observe water storage change. Thus, we argue that it is time to reconsider the possibility of monitoring and eventually closing the water balance even at the event scale, and this is one of the novel aspects of the HPE monitoring approach that we intend to present here. A holistic understanding of the dynamics of an extreme event can only be achieved if all four components of the water balance can be assessed and brought into balance.

We totally agree with the reviewer's opinion that the analysis is stronger if only observations were used for the water balance analysis. This is exactly what we do in the present analysis. We actually did not use ERA5 for quantifying precipitation and just rely on our high resolution observations. We used ERA5 only for comparison to our measurements of atmospheric water vapour in Figure 5 and to

determine the origin of the moisture transported to the observational region (Figure 6). In the schematic diagram of Figure 2 we included ERA5 as data source for completeness, but as it is not used for the water balance analysis in our study this seems to be misleading and it will be removed from Figure 2 in the revised version.

*Finally, I see that this is beyond the control of the authors, but it would, of course, be soo much more interesting if an actual HPE was recorded. If the campaign still continues and the authors do agree to split the manuscript, I would wait with publishing the water balance study untill an HPE has been recorded.*

**Reply**: Due to the MOSES observation concept and the focus on short term event-oriented campaigns, we will not always be able to catch the target events of highest intensity. However, the aim of our paper is to introduce the observation strategy and to show the prove of concept. Meanwhile, the MOSES approach was further established and adapted to other regions and target events. The results of these experiments are going to be submitted in recent publications (e.g. Kunz et al., in prep.).

**4)     Comment on the new gravimetry method**

*Section 3.5 on the gravimetric sensor should report on the usabillity of this setup for hydrology in general. It is currently presented with the results, but mainly describes the sensor and setup, so should be in methods. I'm keen to see this device and what it can do!*

**Reply:** In the course of restructuring the manuscript as mentioned above by moving the technical sensor and setup details to the appendix, we will keep a short paragraph on the overall potential of the gravimetric monitoring in the introductory section of the selected observation techniques in main text. As mentioned in the manuscript, due to the test character of the presented campaigns and technical failures we were unfortunately not able to record usable science data from the gravimetric monitoring campaigns in the Müglitz basin. First results on the usability of the gravimetric setup for hydrology has recently be shown for another study area in Heisterkamp et al. 2022, and the general usability in hydrology has been shown in Güntner et al. (2017), albeit with another gravimeter type, and in other publications, partly also cited in the present manuscript.

Heistermann, M., Bogena, H., Francke, T., Güntner, A., Jakobi, J., Rasche, D., Schrön, M., Döpper, V., Fersch, B., Groh, J., Patil, A., Pütz, T., Reich, M., Zacharias, S., Zengerle, C., Oswald, S. (2022): Soil moisture observation in a forested headwater catchment: combining a dense cosmic-ray neutron sensor network with roving and hydrogravimetry at the TERENO site Wüstebach. - Earth System Science Data, 14, 5, 2501-2519. https://doi.org/10.5194/essd-14-2501-2022

Güntner, A., Reich, M., Mikolaj, M., Creutzfeldt, B., Schröder, S., Wziontek, H. (2017): Landscape-scale water balance monitoring with an iGrav superconducting gravimeter in a field enclosure. - Hydrology and Earth System Sciences, 21, 6, 3167-3182. https://doi.org/10.5194/hess-21-3167-2017